# Differentially Private and Scalable Estimation of the Network Principal Component

**Alireza Khayatian**                                                    *alireza.khayatian@kuleuven.be*
*ESAT-STADIUS*
*KU Leuven*

**Anil Vullikanti**                                                          *asv9v@virginia.edu*
*Department of Computer Science*
*University of Virginia*

**Aritra Konar**                                                            *aritra.konar@kuleuven.be*
*ESAT-STADIUS*
*KU Leuven*

**Reviewed on OpenReview:** *https://openreview.net/forum?id=VOBjWbrAYC*

## Abstract

Computing the principal component (PC) of the adjacency matrix of an undirected graph has several applications ranging from identifying key vertices for influence maximization and controlling diffusion processes, to discovering densely interconnected vertex subsets. However, many networked datasets are sensitive, which necessitates private computation of the PC for use in the aforementioned applications. Differential privacy has emerged as the gold standard in privacy-preserving data analysis, but existing DP algorithms for private PC suffer from low accuracy due to large noise injection or high complexity. Motivated by the large gap between the local and global sensitivities of the PC on real-graphs, we consider instance-specific mechanisms for privately computing the PC under edge-DP. These mechanisms guarantee privacy for all datasets, but provide good utility on "well-behaved" datasets by injecting smaller amounts of noise. More specifically, we consider the Propose-Test-Release (PTR) framework. Although computationally expensive in general, we design a novel approach for implementing a PTR variant in the same time as computation of a non-private PC, while offering good utility. Our framework tests in a differentially-private manner whether a given graph is "well-behaved" or not, and then tests whether its private to release a noisy PC with small noise. As a consequence, this also leads to the first DP algorithm for the Densest-$k$-subgraph problem, a key graph mining primitive. We run our method on diverse real-world networks, with the largest having 3 million vertices, and compare its utility to a pre-existing baseline based on the private power method (PPM). Although PTR requires a slightly larger privacy budget, on average, it achieves a 180-fold improvement in runtime over PPM.

## 1 Introduction

**Background:** The principal component **v** of a network, i.e., the eigen-vector corresponding to the largest eigen-value of the graph adjacency matrix (and more generally, the other principal components), has been found useful in diverse tasks in graph mining Bonacich (2007); Das et al. (2018); Le et al. (2015). The component $v_i$ corresponding to node $i$, is commonly referred to as its eigen-vector centrality. This notion has been used to identify critical nodes in many types of networks, e.g., social and biological networks Lohmann et al. (2010); Jalili et al. (2016); Das et al. (2018). It has been shown that nodes with high eigen-vector centrality are good solutions for influence maximization Dey et al. (2019); Maharani et al. (2014); Deng et al.

(2017). Interestingly, vaccination of high eigen-vector centrality nodes has also been found to be effective in reducing diffusion processes Van Mieghem et al. (2011); Saha et al. (2015); Doostmohammadian et al. (2020). More generally, eigen-vector centrality is used as a standard baseline in most analyses where a set of well connected nodes need to be selected.

Another application of the principal component is in identifying highly interconnected subsets of vertices in a graph, which is a fundamental problem in graph mining with wide-ranging applications spanning bioinformatics, social network analysis, and fraud detection (see Lanciano et al. (2024) and references therein). The densest-$k$-subgraph (D$k$S) Feige et al. (2001) corresponds to a vertex subset of pre-specified size $k$ with the largest (induced) edge density in a graph. The D$k$S problem falls within the broader class of density-based subgraph detection techniques, which include popular measures such as the core decomposition Seidman (1983) and the densest subgraph problem Charikar (2000); Goldberg (1984). In contrast to these approaches, the D$k$S formulation possesses the built-in feature of explicit control over the size of the selected subgraph. This is an important advantage enjoyed by D$k$S, as the lack of size control causes the latter approaches to output large subgraphs that are sparsely interconnected in real-world graphs Tsourakakis et al. (2013); Shin et al. (2016). On the flip-side, the D$k$S problem is computationally intractable in then worst-case, and achieving good approximation guarantees is provably hard under standard complexity-theoretic assumptions Manurangsi (2017); Jones et al. (2023). Notwithstanding these negative results, the work of Papailiopoulos et al. (2014) demonstrated that employing a low-rank approximation of the adjacency matrix based on its principal components can work well in practice, and also provides data dependent approximation guarantees. In particular, it was demonstrated that simply using a rank-1 approximation based on the principal component **v** can provide effective approximation for D$k$S.

**Differential privacy:** In many applications such as public health, social networks, and finance, network datasets comprise sensitive information, and re-identification and other kinds of privacy risks are serious issues Yuan et al. (2024); Peng et al. (2012). For instance, consider a social network where the identities of the vertices (i.e., individuals) are public, but the edges (i.e., personal contacts) are private. However, merely concealing the link structure is insufficient to ensure privacy, as an adversary can still infer the presence or absence of specific edges by analyzing query outputs from the dataset Jiang et al. (2021). Although several privacy models have been proposed, Differential Privacy (DP) Dwork et al. (2014a; 2006b) has emerged as a powerful and broadly adopted framework for developing algorithms that offer rigorous, quantifiable privacy guarantees without making assumptions about the adversary's knowledge or capabilities. In the context of graphs, there are two standard notions of privacy - edge privacy, where vertices are public and edges are private. In this case, the objective is to prevent the disclosure of the existence/non-existence of an arbitrary edge in the input graph. On the other hand, in node privacy Kasiviswanathan et al. (2013); Blocki et al. (2013), vertices are also private, and the goal is to protect any arbitrary vertex and its incident edges. DP algorithms have been developed for many graph problems, ranging from clustering and community detection, Epasto et al. (2022a); Mohamed et al. (2022); Nguyen & Vullikanti (2024), to personalized PageRank Epasto et al. (2022b); Wei et al. (2024), and computing different kinds of graph statistics Karwa et al. (2014); Gupta et al. (2012); Blocki et al. (2013).

There has also been work on computing principal components under edge DP Kapralov & Talwar (2013); Chaudhuri et al. (2012); Hardt & Roth (2013); Hardt & Price (2014); Balcan et al. (2016); Gonem & Gilad-Bachrach (2018). However, as we discuss in Section 2, prior methods either do not have adequate accuracy, or do not scale well even to networks of moderate size. This is the main motivation of our work. We also note that the D$k$S problem has not been studied under DP.

**Contributions:** In this paper, we develop a scalable algorithm for differentially private computation of the principal component (PC) of the graph adjacency matrix under edge-DP. This also results in the first edge-DP algorithm for the D$k$S, leveraging the non-private low-rank approximation approach of Papailiopoulos et al. (2014). DP algorithms achieve their privacy guarantees by adding controlled amounts of noise to non-private computations. While algorithms for privately computing PCs of (general) matrices are known, as mentioned above, one issue is that the DP guarantees provided by these methods are based on worst-case outcomes across *all* possible datasets. This can result in adding excessive noise, which has a detrimental effect on utility, since a given dataset need not be representative of the worst-case. To address this problem, the work of Gonem & Gilad-Bachrach (2018) developed a technique for privately computing PCs for "well-behaved" datasets

that inject smaller amounts of noise via the smooth sensitivity framework of Nissim et al. (2007). Such instance-specific mechanisms are appealing since they are private for every dataset, but can offer improved accuracy on "well-behaved" datasets. Our work also concerns the use of instance-specific mechanisms for computing the PC under edge-DP, albeit we use the Propose-Test-Release (PTR) framework Dwork & Lei (2009), since smooth sensitivity gives poor results in our context (see further below). In general, PTR is not "user-friendly" as it involves computations which are often challenging to implement in polynomial-time. Hence, the main contribution of our work is to develop a scalable and practical variant of PTR for private PC. Our contributions can be further summarized below.

**(1)** Our approach is based on *output perturbation*, where the principal component (PC) of the adjacency matrix is first computed non-privately followed by a one-shot noise addition step to provide DP. First, we derive a new $\ell_2$ local sensitivity bound under edge DP (Theorem 1), which we then use to highlight the large gap between the local and global sensitivities of the PC on real-world graphs. This implies that the standard approach of calibrating noise to global sensitivity Dwork et al. (2014a) leads to poor accuracy see figure 1), motivating the application of instance-based mechanisms such as smooth sensitivity Nissim et al. (2007) or Propose-Test-Release Dwork & Lei (2009). These mechanisms calibrate noise to local sensitivity-based estimates to provide DP, thereby offering improved privacy-utility trade-offs. However, a key challenge is that they are difficult to implement in practice, as they need not involve polynomial-time computations in general.

**(2)** We then derive a tight analysis of prior bounds on computing private PC via smooth sensitivity Gonem & Gilad-Bachrach (2018) (developed to address the high computational complexity of Nissim et al. (2007)), and show that these are very close to the global sensitivity value, in general (see Appendix D).

**(3)** Given the unsuitability of smooth sensitivity, we shift our focus to using the Propose-Test-Release (PTR) framework Dwork & Lei (2009); Li et al. (2024). At a high level, PTR requires proposing a bound $\beta$ for the local sensitivity of the PC on the given dataset, followed by a differentially private test to see if adding noise to the PC calibrated to $\beta$ would violate privacy. However, PTR is computationally very expensive in general, and the only prior work for implementing PTR for graph problems in polynomial time was Li et al. (2023) for an unrelated problem of computing epidemic metrics. Our main contribution is to introduce a computationally efficient and practical PTR variant for private PC (See Figure 2 for a flow diagram). The algorithm proceeds in three phases. In the first phase, it features a differentially-private test to filter out graphs which are not "well-behaved" (i.e., exceed a fixed spectral gap threshold) without resulting in false positives via application of the Truncated Biased Laplace (TBL) mechanism Xiao et al. (2025). In the second phase, a private test is performed to compute a test statistic $\hat{\phi}(\mathcal{G})$ that constitutes a lower bound on the distance to instances with local sensitivity exceeding a proposed bound $\beta$, based on a novel technique (see Theorem 2), and a more efficient procedure for selecting the parameter $\beta$ (see Theorem 5 and Proposition 1). In the third and final phase, $\hat{\phi}(\mathcal{G})$ is compared against a threshold to decide privately whether the noisy principal component should be released or not.

*Our methods reduce the complexity of PTR for private PC to basically the same as computation of the PC, which gives us significant improvement in terms of running time.*

We also employ the iterative private power method (PPM) of Hardt & Price (2014) as a baseline for comparison. Through experiments on real graphs, we provide compare and contrast the efficacy of these two approaches for two applications: **(A1)** privately extracting the subset with the top-$k$ eigenscores, and **(A2)** private approximate D$k$S. A representative example for D$k$S is provided in Figure 1, which depicts the edge-density (whihc measures utility) versus size trade-off on a social network with 3 million vertices and 120 million edges. The yellow line is the rank-1 non-private algorithm of Papailiopoulos et al. (2014), whereas the red and blue lines depict PTR and PPM respectively. Both algorithms perform comparably to the non-private version (PPM offers slightly better privacy and accuracy). However, PTR is 700 times faster than PPM as it's a one-shot noise addition mechanism.

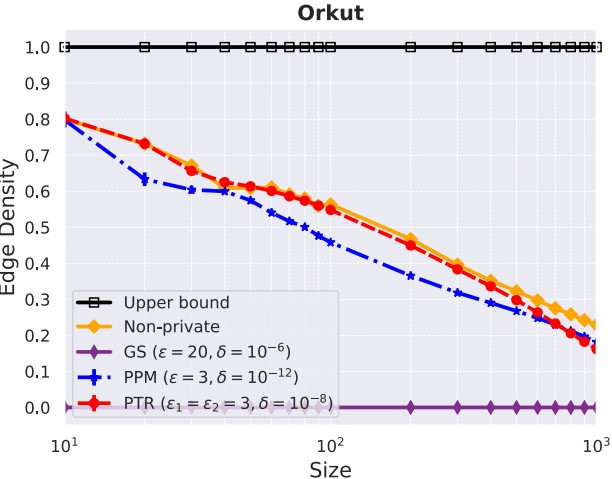

Figure 1: Comparison of the edge-density (y-axis) versus subgraph size (x-axis) trade-off on the Orkut social network with $3M$ vertices and $120M$ edges. PPM (blue) and PTR (red) $((\epsilon_0, \delta_0) = (1, 7e^{-7}))$ attain performance comparable with their non-private counterpart (yellow). However, PTR is $\approx 700$ times faster than PPM. The standard Gaussian mechanism with noise calibrated to global sensitivity (purple) yields poor results.

## 2 Related Work

We briefly summarize relevant related work here. We first discuss the use of nodes with high eigen-vector centrality in different applications, and the D$k$S problem, and then the problems of differential privacy for graph problems and private principal component analysis (PCA).

**Eigen-vector centrality and applications:** Centrality metrics are common tools used in network analysis. The eigen-vector centrality (also referred to as eigenscore) of node $i$ is defined as the $i$th component of the principal component $\mathbf{v}$ of $G$ Bonacich (2007); Das et al. (2018); Le et al. (2015); this notion has been found very effective in a number of applications. For instance, nodes with high eigen-vector centrality have been found to be effective for controlling diffusion processes on networks, both for maximizing influence or information spread Dey et al. (2019); Maharani et al. (2014); Deng et al. (2017) and for controlling the spread of epidemic processes by choosing such nodes for vaccination interventions Van Mieghem et al. (2011); Saha et al. (2015); Doostmohammadian et al. (2020). Eigen-vector centrality has also been used in other domains such as biological networks for identifying essential proteins and critical components Lohmann et al. (2010); Jalili et al. (2016).

**The D$k$S problem:** The D$k$S problem is computationally very hard, and it has been shown that an $O(n^{1/(\log\log n)^c})$ approximation is not possible under certain complexity theoretic assumptions Manurangsi (2017) (see other discussion on its complexity in Khot (2006); Manurangsi (2017); Bhaskara et al. (2012)). Efficient algorithms which work well for practical instances of D$k$S range from low-rank approximations Papailiopoulos et al. (2014) and the convex and non-convex relaxations of Konar & Sidiropoulos (2021) and Lu et al. (2025). We mention in passing that there are variants of D$k$S, such as the Densest at-least-$k$ Subgraph problem (Dal$k$S), the Densest at-most-$k$ Subgraph problem (Dam$k$S) Andersen & Chellapilla (2009); Khuller & Saha (2009), and the $f$-densest subgraph problem Kawase & Miyauchi (2018). Although these variants impose size constraints on the extracted subgraph, these formulations do not guarantee that the entire spectrum of densest subgraphs (i.e., of every size) can be explored.

**Differential Privacy for Graphs:** In the context of graph datasets, there are two standard notions of privacy - edge privacy Blocki et al. (2013), where vertices are public and edges are private. In this case, the objective is to prevent the disclosure of the existence/non-existence of an arbitrary edge in the input graph.

On the other hand, in node privacy Kasiviswanathan et al. (2013), vertices are also private, and the goal is to protect any arbitrary vertex and its incident edges.

Early work on differentially private graph algorithms focused on computing basic statistics. Nissim et al. introduced DP for graph computations, providing methods to privately release the cost of minimum spanning trees and triangle counts using smooth sensitivity Nissim et al. (2007). Karwa et al. Karwa et al. (2014) extended these techniques to other subgraph structures, such as k-stars and k-triangles. Hay et al. Hay et al. (2009) explored DP mechanisms for releasing degree distributions, while Gupta et al. Gupta et al. (2012) developed methods for privately answering cut queries. Node privacy, being more sensitive, is more challenging to ensure. Kasiviswanathan et al. Kasiviswanathan et al. (2013) and Blocki et al. Blocki et al. (2013) addressed this issue by developing node-DP algorithms using Lipschitz extensions for subgraph counting.

Recently, two concurrent works Farhadi et al. (2022); Nguyen & Vullikanti (2021) introduced edge DP algorithms for computing the densest subgraph (DSG), which aims to find the subgraph that maximizes the average induced degree Goldberg (1984). Building on the non-private greedy peeling algorithm of Charikar (2000) for DSG, Nguyen et al. Nguyen & Vullikanti (2021) employed the exponential mechanism Dwork et al. (2014a) to design an $(\epsilon, \delta)$-DP algorithm which can be executed in both sequential and parallel versions. Meanwhile, Farhadi et al. Farhadi et al. (2022) applied the Prefix Sum Mechanism Chan et al. (2011) to privatize Charikar's greedy peeling algorithm for DSG, resulting in a pure $(\epsilon, 0)$-DP algorithm that runs in linear time. Most recently, the work of Dinitz et al. (2025) proposed an $(\epsilon, \delta)$-DP algorithm that is based on using a private variant of the multiplicative weights framework Arora et al. (2012) for solving a tight linear programming relaxation of DSG proposed in Chekuri et al. (2022). In a separate but related work, Dhulipala et al. Dhulipala et al. (2022) develop algorithms for releasing core numbers with DP, which are generated using a variant of the greedy peeling algorithm. At present, we are unaware of any DP algorithm for the D$k$S problem.

**Private Principal Component Analysis (PCA):** The topic of differentially private PCA has been explored previously in various studies, including Chaudhuri et al. (2012); Hardt & Roth (2013); Kapralov & Talwar (2013); Blum et al. (2005); Dwork et al. (2014b). Initially, Blum et al. (2005); Dwork et al. (2014b) considered an general input perturbation approach for providing DP by adding random Gaussian noise to the empirical covariance matrix constructed from the data points. However, this method requires access to the covariance matrix, which is computationally expensive in both space and time. Although graph adjacency matrices are not covariances (as they may not be positive semi-definite), the classic randomized response technique of Warner Warner (1965) can be applied to generate DP adjacency matrices, with the caveat that it can result in injection of large amount of noise, which manifests in creating dense graphs with very different properties from the actual dataset. This can result in poor utility for node subset selection problems (see (Farhadi et al., 2022, Appendix A)).

An alternative approach based on output perturbation that utilizes the exponential mechanism to output principal components with $(\epsilon, 0)$-DP was explored in Kapralov & Talwar (2013); Chaudhuri et al. (2012). However, the theoretical analysis of this method shows that it incurs a high time complexity of $O(n^6)$ (here $n$ represents the dimension), which renders it impractical for application on even moderately sized datasets. Meanwhile, the work of Hardt & Roth (2013); Hardt & Price (2014); Balcan et al. (2016) consider privatizing power iterations for computing private subspaces with DP. This iterative approach comprises of interleaved steps of matrix vector multiplication followed by noise injection and re-normalization to provide $(\epsilon, \delta)$-DP. While it can perform well in practice, selecting its parameters requires some trial-and-error and its iterative nature can result in high-complexity when applied to large datasets. Closest to our present work is the approach of Gonem & Gilad-Bachrach (2018), which considered using the smooth sensitivity framework of Nissim et al. (2007) to privately compute principal components via output perturbation. However, in practice, smooth sensitivity often fails to reduce the noise sufficiently. We address this issue by designing a method to significantly reduce the noise level in output perturbation using the propose-test-release (PTR) framework.

## 3  Preliminaries

**Differential Privacy on Graphs:** Let $\mathbb{G}$ denote a collection of undirected graphs on a fixed set $\mathcal{V}$ of nodes. This paper focuses on the notion of edge privacy Blocki et al. (2013), where two graphs $\mathcal{G}, \mathcal{G}' \in \mathbb{G}$

are considered neighbors, denoted $\mathcal{G} \sim \mathcal{G}'$, if they have the same vertex set and differ in exactly one edge, meaning $|\mathcal{E}(\mathcal{G}) - \mathcal{E}(\mathcal{G}')| = 1$.

**Definition 1.** *$(\epsilon, \delta)$-Differential Privacy (DP) Dwork et al. (2014a). Let $\epsilon > 0$ and $\delta \in [0, 1]$. A randomized algorithm $\mathcal{A}(\cdot) : \mathbb{G} \to \mathbb{R}^k$ is $(\epsilon, \delta)$-DP if:*

$$\Pr(\mathcal{A}(\mathcal{G}) \in S) \leq \exp(\epsilon) \Pr(\mathcal{A}(\mathcal{G}') \in S) + \delta, \tag{1}$$

*for any outcome $S \subseteq \mathbb{R}^k$, and for any pair of neighboring datasets $(\mathcal{G}, \mathcal{G}')$ that differ in a single edge.*

**Function Sensitivity:** Dwork et al. (2014a). Consider a function $f : \mathbb{G} \to \mathbb{R}^k$. The *local $\ell_2$ sensitivity* of $f$ exhibited by a dataset $\mathcal{G} \in \mathbb{G}$ is defined as $\mathsf{LS}_f(\mathcal{G}) = \max_{\mathcal{G}':\mathcal{G}\sim\mathcal{G}'} \|f(\mathcal{G}) - f(\mathcal{G}')\|_2$, where the maximum is taken over all neighboring datasets $\mathcal{G}'$ which differ in one edge from $\mathcal{G}$. The *global $\ell_2$ sensitivity* of $f$ is then $\mathsf{GS}_f = \max_{\mathcal{G}\in\mathbb{G}} \mathsf{LS}_f(\mathcal{G})$. If we computed the sensitivity calculations w.r.t. the $\ell_1$ norm instead, we would obtain the $\ell_1$ local and global sensitivities of $f$. Adding i.i.d. Gaussian noise to $f(\cdot)$ with variance appropriately calibrated to global $\ell_2$ sensitivity is guaranteed to satisfy $(\epsilon, \delta)$-DP Dwork et al. (2014a). By adding Laplacian noise calibrated to global $\ell_1$ sensitivity, the output is guaranteed to be $(\epsilon, 0)$-DP Dwork et al. (2014a). An important property of DP is that applying post-processing on the output of a $(\epsilon, \delta)$-DP mechanism does not weaken the privacy guarantee.

**Private principal components:** Consider an unweighted, undirected simple graph $\mathcal{G} := (\mathcal{V}, \mathcal{E})$ with vertex set $\mathcal{V} := \{1, \ldots, n\}$ and edge set $\mathcal{E} \subseteq \mathcal{V} \times \mathcal{V}$. The $n \times n$ symmetric adjacency matrix of $\mathcal{G}$ is denoted as $\mathbf{A}$. The eigen-values of $\mathbf{A}$ are arranged in non-increasing order of their magnitudes; i.e., we have $|\lambda_1| \geq |\lambda_2| \geq \cdots |\lambda_n|$. The principal eigen-gap of $\mathcal{G}$ is defined as $\mathsf{GAP}(\mathcal{G}) := |\lambda_1| - |\lambda_2|$, while the principal component $\mathbf{v}$ of $\mathbf{A}$ is the eigen-vector associated with $|\lambda_1|$. If the graph $\mathcal{G}$ represented by $\mathbf{A}$ is connected, the Perron-Frobenius Theorem Meyer (2023) asserts that $\mathbf{v}$ is element-wise positive and $\lambda_1 > 0$. Following our earlier notation, we consider the function $f(\mathcal{G}) = \mathbf{v}$, and $\mathsf{LS}_\mathbf{v}(\mathcal{G}) = \max_{\mathcal{G}':\mathcal{G}\sim\mathcal{G}'} \|\mathbf{v} - \mathbf{v}'\|_2$ In this paper, we develop new techniques for privatizing $\mathbf{v}$ under edge-DP, with the following two applications.

**(A1) Eigen-vector centrality:** For a node $i$, its component $v_i$ in the principal component $\mathbf{v}$ is referred to as its eigen-vector centrality or eigenscore. Nodes with high eigen-vector centrality (i.e., the $k$-largest entries of $\mathbf{v}$) have been used in a number of applications, such as identifying influential users in social networks and for targeted interventions for controlling epidemic processes, e.g., Maharani et al. (2014); Van Mieghem et al. (2011); Saha et al. (2015). Here we study how to find the $k$-largest entries of $\mathbf{v}$ with edge-DP.

**(A2) Densest-$k$-Subgraph:** Consider a vertex subset $\mathcal{S} \subseteq \mathcal{V}$ which induces a subgraph $\mathcal{G}_\mathcal{S} = (\mathcal{S}, \mathcal{E}_\mathcal{S})$, where $\mathcal{E}_\mathcal{S} = \{\{u, v\} \in \mathcal{E} \mid u, v \in \mathcal{S}\}$. The edge density of the subgraph $\mathcal{G}_\mathcal{S}$ is defined as $d(\mathcal{G}_\mathcal{S}) = |\mathcal{E}_\mathcal{S}|/\binom{|\mathcal{S}|}{2}$, which measures what fraction of edges in $\mathcal{G}_\mathcal{S}$ are connected. Note that larger values correspond to dense quasi-cliques, with a maximum value of 1 for cliques. The Densest-$k$-Subgraph (D$k$S) problem Feige et al. (2001) seeks to find a subset of $k$ vertices $\mathcal{S} \subseteq \mathcal{V}$ that maximizes the edge density. Formally, the problem can be expressed as follows

$$d_k^* := \max_{\mathbf{x} \in \mathcal{X}_k} \left\{ \frac{\mathbf{x}^T \mathbf{A} \mathbf{x}}{\binom{k}{2}} \right\} \tag{2}$$

where $\mathcal{X}_k := \{\mathbf{x} \in \{0, 1\}^n : \mathbf{e}^T \mathbf{x} = k\}$ represents the combinatorial selection constraints. Here, each binary vector $\mathbf{x} \in \{0, 1\}^n$ represents a vertex subset $\mathcal{S} \subseteq \mathcal{V}$, with $x_i = 1$ if $i \in \mathcal{S}$ and $x_i = 0$ otherwise.

## 4  Analyzing sensitivity of principal components

In order to compute the principal component of $\mathbf{A}$ under edge-DP , a straightforward idea is to apply the technique of *output perturbation.* This can be achieved by computing the global sensitivity of $\mathbf{v}$ and then applying the Gaussian mechanism to provide $(\epsilon, \delta)$ DP Dwork et al. (2006a). Since output perturbation is a one-shot noise addition scheme, it is computationally fast, as it incurs only $O(n)$ complexity. However, the global $\ell_2$ sensitivity of $\mathbf{v}$ is large; at most $\sqrt{2}$ (Gonem & Gilad-Bachrach, 2018, Theorem 5). This implies that a large amount of noise has to be injected to achieve a target privacy requirement, which can result in a severe degradation in utility. On the other hand, the local $\ell_2$ sensitivity can be substantially smaller on many graphs. Note that adding noise calibrated to the local sensitivity is not guaranteed to be privacy preserving

Table 1: Comparison between local and global sensitivity for real-world datasets.

| Graph | $n$ | GAP($\mathcal{G}$) | LS$_\mathbf{v}$ | GS$_\mathbf{v}$/LS$_\mathbf{v}$ |
|---|---|---|---|---|
| Facebook | 4k | 36.9 | 7e-3 | 202 |
| PPI-Human | 21k | 70.8 | 4e-4 | 3535 |
| soc-BlogCatalog | 89k | 335.9 | 7e-4 | 2035 |
| Flickr | 105k | 101.3 | 1e-3 | 1414 |
| Twitch-Gamers | 168k | 148.4 | 2e-3 | 539 |
| Orkut | 3M | 225.4 | 1e-2 | 141 |

Nissim et al. (2007). However, if we observe that the local $\ell_2$ sensitivity of $\mathbf{v}$ is much smaller than $\sqrt{2}$ on a given graph, we can employ *instance-specific* noise addition schemes Nissim et al. (2007); Dwork & Lei (2009), which provide DP by injecting noise calibrated noise to local sensitivity calculations. We demonstrate that under a mild requirement on the spectral gap of $\mathcal{G}$, the following bound on the local sensitivity can be derived.

**Theorem 1.** *If GAP($\mathcal{G}$) > $\sqrt{2}(\sqrt{2}+1)$, then the local $\ell_2$ sensitivity of $\mathbf{v}$ under edge-DP is at most*

$$LS_\mathbf{v}(\mathcal{G}) \leq \frac{2\sqrt{v_{\pi(1)}^2 + v_{\pi(2)}^2}}{GAP(\mathcal{G})} = \frac{2c_\pi}{GAP(\mathcal{G})}, \tag{3}$$

*where $c_\pi = \sqrt{v_{\pi(1)}^2 + v_{\pi(2)}^2}$ and $v_{\pi(1)}$ and $v_{\pi(2)}$ denote the largest and second-largest entries of $\mathbf{v}$ respectively.*

*Proof.* Please refer to Appendix B. $\square$

Note that the numerator term in the above upper bound is at most $2\sqrt{2}\|\mathbf{v}\|_\infty$. We conclude that for graphs whose eigen-gap GAP($\mathcal{G}$) exceeds $\sqrt{2}(\sqrt{2}+1)$, the local $\ell_2$ sensitivity of the principal component $\mathbf{v}$ is small when the entries of $\mathbf{v}$ are "spread out" in magnitude (i.e., $\|\mathbf{v}\|_\infty$ is small ) and the eigen-gap is large.

The value of this estimate for various real-world graph datasets is presented in Table 1, and contrasted with the global sensitivity upper bound of $\sqrt{2}$. It is evident that the local sensitivity of $\mathbf{v}$ on real graphs can be at least 2 orders of magnitude smaller compared to the global sensitivity estimate of $\sqrt{2}$. This motivates the application of instance-specific noise-addition mechanisms for providing DP on such "good" instances, the first of which we consider is the smooth sensitivity framework of Nissim et al. (2007). However, practical application of this framework is challenging, as it entails computations which need not be polynomial-time. The prior work of Gonem & Gilad-Bachrach (2018) developed tractable smooth upper bounds on the smooth sensitivity for computing principal components of general datasets; albeit not for graphs under edge-DP. In Appendix D, we provide a rigorous analysis which shows that in our graph setting under edge-DP, the obtained smooth upper bound is very close to the global sensitivity estimate of $\sqrt{2}$.

## 5 The Propose-Test-Release Mechanism

Since adopting the smooth sensitivity-based approach of Gonem & Gilad-Bachrach (2018) does not yield tangible improvements over the global sensitivity, we explore the application of an alternative instance-specific noise injection mechanism; namely, the Propose-Test-Release (PTR) framework introduced in Dwork & Lei (2009). While PTR and smooth sensitivity share a common aim (i.e., exploiting local sensitivity to add instance-specific noise while preserving DP), PTR is a distinct paradigm, which presents its own unique challenges.

At a high level, the PTR framework comprises the following steps. Given an upper bound $\beta$ on the local $\ell_2$ sensitivity LS$_\mathbf{v}(\mathcal{G})$, test (in a differentially private manner) whether the current dataset is "close" to another with high sensitivity. If so, the algorithm can refuse to yield a response; otherwise, it can release a private principal component with a small amount of noise (scaled to $\beta$). The main difficulty in implementing PTR

lies in the "test" component, which requires computing the Hamming distance to the nearest dataset $\mathcal{G}'$ whose local sensitivity $\mathsf{LS_v}(\mathcal{G}')$ exceeds $\beta$. Computing such a sensitivity-1 statistic requires solving the following problem

$$\gamma(G) =: \left\{ \min_{\mathcal{G}'} d(\mathcal{G}, \mathcal{G}') \quad \text{s.to} \quad \mathsf{LS_v}(\mathcal{G}') \geq \beta \right\}, \tag{4}$$

which is difficult in general. In a recent breakthrough, it was shown in Li et al. (2024) that the PTR framework can still be successfully applied if $\gamma(\mathcal{G})$ is replaced by any non-negative, sensitivity-1 statistic $\phi(\mathcal{G})$ that is a global lower bound on $\gamma(\mathcal{G})$; i.e., we have

$$\gamma(\mathcal{G}) \geq \phi(\mathcal{G}) \geq 0, \forall\, \mathcal{G}. \tag{5}$$

This results in the following modified PTR framework; first introduced in Li et al. (2024).

1. Propose an upper bound $\beta$ on the local $\ell_2$ sensitivity $\mathsf{LS_v}(\mathcal{G})$.

2. Compute lower bound $\phi(\mathcal{G})$ on the distance $\gamma(\mathcal{G})$ defined in problem equation 4.

3. Release $\hat{\phi}(\mathcal{G}) := \phi(\mathcal{G}) + \mathrm{Lap}\left(\frac{1}{\epsilon_1}\right)$.

4. If $\hat{\phi}(\mathcal{G}) \leq \frac{\ln(1/\delta)}{\epsilon_1}$, return no response.

5. If $\hat{\phi}(\mathcal{G}) \geq \frac{\ln(1/\delta)}{\epsilon_1}$, return $\mathbf{v} + \mathcal{N}\left(0, \sigma^2 \cdot \mathbf{I}_n\right)$, where $\sigma^2 = 2\beta^2 \log(2/\delta)/\epsilon_2^2$

The upshot is that *if* we can find a suitable surrogate function $\phi(\cdot)$ which satisfies equation 5, and is simpler to compute compared to solving equation 4 for $\gamma(\cdot)$, then we can apply the framework successfully. It is important to note that $\beta$ need not be private for the correctness of the above technique to go through. The work of Li et al. (2024) shows that the output of the modified-PTR algorithm is $(\epsilon_1 + \epsilon_2, \delta)$-DP. However, it does not provide a general recipe for computing such requisite surrogates. Hence, a major contribution of our work lies in constructing a suitable $\phi(\cdot)$ for the given problem. Below, we provide an overview of our approach (see Figure 2 for a flow-diagram).

**Overview of Algorithm 1:** At a high level, our algorithm privately releases the principal component of a graph using a one-shot output perturbation scheme, while avoiding the excessive noise required by worst-case global sensitivity bounds. The main idea is to test whether the input graph is sufficiently distant from other instances with high local sensitivity in a differentially private manner, and to use local sensitivity-based calculations to release a noisy eigenvector only if this test succeeds. Concretely, the algorithm can be broken down into three phases, as illustrated in Figure 2:

*Phase I*: A private *gap test* is carried out to determine whether the graph lies in a "well-behaved" regime with sufficiently large spectral gap.

*Phase II*: Depending on the outcome of Phase I, a follow-up *distance-to-instability test* is performed to check whether the graph is far from instances with large local sensitivity. If the gap test fails, (i.e., the algorithm determines that the graph lies in the small-gap regime), then it treats the graph as unstable, with 0 distance to instability. On the other hand, if the gap test succeeds, the algorithm performs a conservative, but tractable test to lower bound the true distance to unstable instances.

*Phase III*: A private check is performed via the Laplace mechanism to determine whether the computed distance exceeds a fixed threshold. If the test fails, the algorithm does not release a response w.h.p. Otherwise, it releases a private estimate of the principal component by adding Gaussian noise calibrated to a data-dependent sensitivity bound.

Next, we formalize the main ideas underpinning our approach.

● **Construction of $\phi(\cdot)$:** Our goal is to construct a surrogate $\phi(\cdot)$ which satisfies equation 5 for every graph $\mathcal{G}$. For this task, we consider two regimes: (a) a regime where the eigen-gap of $\mathcal{G}$ is "large enough" and (b) a complementary regime where the eigen-gap is "small". We consider each regime separately, and make these notions concrete. First, we show that for case (a), the local $\ell_2$ sensitivity of $\mathbf{v}$ on $\mathcal{G}'$ can be appropriately upper bounded in "simple" terms. Formally, we have the following claim.

**Theorem 2.** *Assume that the following two conditions hold:*

**(A1)** $\mathsf{GAP}(\mathcal{G}) > 2(\sqrt{2} + 1)$, **(A2)** $d(\mathcal{G}, \mathcal{G}') < \left(1 - \frac{1}{\sqrt{2}}\right) \cdot \mathsf{GAP}(\mathcal{G})$.

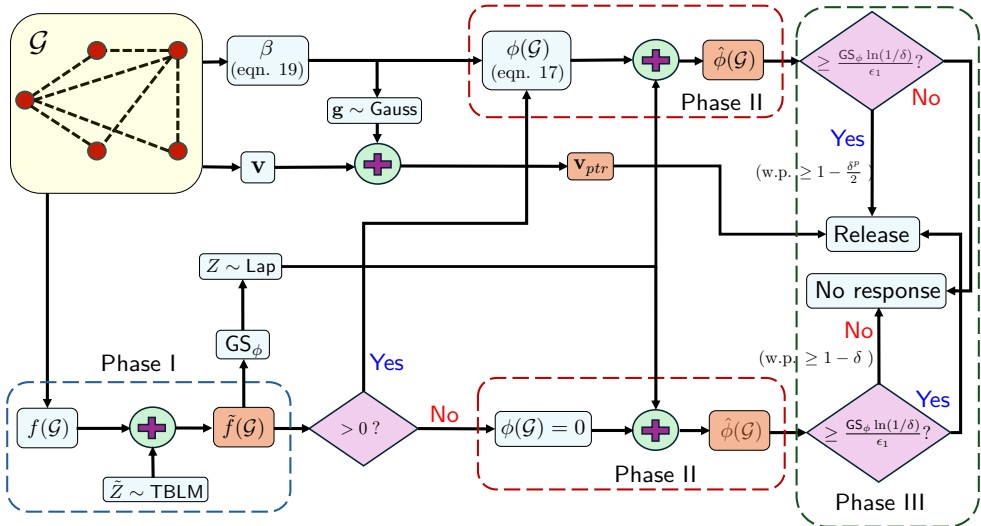

Figure 2: Flow diagram of the proposed PTR algorithm 1. *Input:* Undirected graph $\mathcal{G}$. Under edge-DP, the edges are private but the identities of the vertices are public. The algorithm proceeds in 3 phases, starting from bottom left. *Phase I*: The truncated biased Laplace mechanism (TBLM) is used to privatize the spectral threshold function $f(\mathcal{G}) := \mathsf{GAP}(\mathcal{G}) - t$, where the threshold $t := 2(\sqrt{2}+1)$. The output $\tilde{f}(\mathcal{G})$ is then used to test if $\mathcal{G}$ lies in the large-gap or small-gap regime. *Phase II*: Depending on the outcome of Phase I, the distance to instability $\phi(\mathcal{G})$ is computed and privatized via the Laplace mechanism to obtain $\hat{\phi}(\mathcal{G})$. *Phase III*: The output $\hat{\phi}(\mathcal{G})$ is compared against a threshold to decide privately whether the noisy principal component $\mathbf{v}_{ptr}$ should be released or not.

*Then, the local $\ell_2$ sensitivity of $\mathbf{v}$ on $\mathcal{G}'$ can be upper bounded by*

$$\mathsf{LS}_{\mathbf{v}}(\mathcal{G}') \le \theta(\mathcal{G}') := \frac{2}{\mathsf{GAP}(\mathcal{G}) - d(\mathcal{G}, \mathcal{G}')} \cdot \left[ \frac{2d(\mathcal{G}, \mathcal{G}')}{\mathsf{GAP}(\mathcal{G})} + c_\pi \right]. \tag{6}$$

*Proof.* Please refer to Appendix C. $\square$

Consequently, we can replace $\mathsf{LS}_{\mathbf{v}}(\mathcal{G}')$ in equation 4 with the upper bound $\theta(\mathcal{G}')$. Doing so yields the problem

$$\phi(\mathcal{G}) := \min \; d(\mathcal{G}, \mathcal{G}') \tag{7}$$
$$\text{s.t. } \theta(\mathcal{G}') \ge \beta$$

Since $\theta(G')$ upper bounds $\mathsf{LS}_v(G')$, replacing $\mathsf{LS}_v(G') \ge \beta$ with $\theta(G') \ge \beta$ enlarges the feasible set of equation 4. Hence, it follows that that equation 7 is a relaxation of equation 4, whose solution $\phi(\mathcal{G})$ yields a lower bound on the true PTR distance $\gamma(\mathcal{G})$, as desired. Furthermore, by design, $\phi(\mathcal{G})$ is non-negative and has $\ell_1$ sensitivity equal to 1. Hence, $\phi(\mathcal{G})$ is a suitable candidate for implementing the PTR framework, as it guarantees that equation 5 is satisfied for all graphs for which $\mathsf{GAP}(\mathcal{G}) > 2(\sqrt{2}+1)$, which corresponds to the "sufficiently large" gap regime.

Next, we consider the complementary the small gap regime (b), where $\mathsf{GAP}(\mathcal{G}) \le 2(\sqrt{2}+1)$. Devising a suitable surrogate that obeys equation 5 is more challenging compared to the previous case, since even the local sensitivity bound of Theorem 1 need not apply here. Hence, our goal is to make the algorithm reject such "unfavorable" datasets w.h.p. To this end, we devise a surrogate for $\gamma(\mathcal{G})$ which outputs 0 for such datasets. Clearly, such a surrogate satisfies equation 5. A valid choice is the following optimization problem

$$\min \; d(\mathcal{G}, \mathcal{G}') \tag{8}$$
$$\text{s.t. } \theta(\mathcal{G}') \ge 0.$$

Compared to equation 7, the proposed bound $\beta$ is replaced by 0 in the RHS of the constraint. Regarding the above problem, we have the following claim.

**Lemma 1.** *For every graph $\mathcal{G}$, the optimal value of problem 8 is* 0.

*Proof.* See Appendix E. □

We would like to use equation 8 in PTR only for graphs belonging in the small gap regime $\mathsf{GAP}(\mathcal{G}) \leq 2(\sqrt{2}+1)$. For other "well-behaved" graphs whose gap exceeds the aforementioned threshold, we would like to employ the surrogate equation 7. For this task, we can combine problems equation 7 and equation 8 into the single optimization problem

$$
\begin{aligned}
\phi(\mathcal{G}) := \min\ & d(\mathcal{G}, \mathcal{G}') \\
\text{s.t.}\ & \theta(\mathcal{G}') \geq \beta \cdot u(\mathsf{GAP}(\mathcal{G}) - t),
\end{aligned}
\tag{9}
$$

where $t := 2(\sqrt{2}+1)$ is the gap threshold, and $u(.)$ is the indicator function $u(x) := \mathbf{1}_{\{x>0\}}$. In the large gap regime, problem 9 reduces to 7. Otherwise, it boils down to 8, with value 0. In Appendix H we show that for graphs whose spectral gap is below the threshold $t = 2(\sqrt{2}+1)$, the PTR algorithm is designed to return NO RESPONSE with probability at least $1 - \delta$, as intended. An issue is that instantiating equation 9 first requires checking whether the gap of $\mathcal{G}$ exceeds the threshold $t$ or not, which need not be privacy-preserving. Our proposed solution is to first privatize the function $f(\mathcal{G}) := \mathsf{GAP}(\mathcal{G}) - t$ and then employ it in equation 9. Next, we describe how to perform such an operation.

• **Privatizing $f(\mathcal{G})$ via the Truncated Biased Laplace Mechanism (TBLM):** In order to privatize $f(\mathcal{G})$, we first need to compute the global $\ell_1$ sensitivity of $f(\cdot)$ under edge-DP. A calculation reveals that

$$
\mathsf{GS}_f = \max_{\mathcal{G} \sim \mathcal{G}'} |f(\mathcal{G}) - f(\mathcal{G}')| = \max_{\mathcal{G} \sim \mathcal{G}'} |\mathsf{GAP}(\mathcal{G}) - \mathsf{GAP}(\mathcal{G}')| \leq 1.
\tag{10}
$$

Hence, it suffices to add Laplacian noise $Z \sim \mathsf{Lap}(0, 1/\epsilon_0)$ to $f(\mathcal{G})$ to guarantee $(\epsilon_0, 0)$-DP Dwork et al. (2014a). Denote the noisy output $\tilde{f}(\mathcal{G}) := f(\mathcal{G}) + Z$. However, the standard Laplace mechanism adds two-sided symmetric noise to $f(\mathcal{G})$, which can result in the outcome that $f(\mathcal{G}) \leq 0$, but $\tilde{f}(\mathcal{G}) > 0$. This give rise to undesirable false-positives where we incorrectly employ the surrogate equation 7 instead of equation 8. To prevent such an occurrence, we apply the truncated biased Laplace mechanism (TBLM) Xiao et al. (2025), which adds one-sided noise to provide $(\epsilon_0, \delta_0)$-DP. Formally, given parameters $\mu > 0$, $\lambda > 0$, and $R > 0$, the density of TBL noise is given by

$$
\mathsf{Pr}(Z = z) = \frac{\exp(-|z - \mu|/\lambda)}{Z_{\mu,\lambda,R}} \, \mathbf{1}_{\{0 \leq z \leq R\}},
\tag{11}
$$

where $\mathbf{1}_{\{0 \leq z \leq R\}}$ is the indicator function of the interval $[0, R]$, and $Z_{\mu,\lambda,R} = \mathsf{Pr}(0 \leq z \leq R)$ is the normalization parameter. Note that the samples from the TBL distribution are guaranteed to be positive and bounded. Furthermore, generating such samples can be accomplished efficiently (see Appendix F).

**Fact 1.** *(Xiao et al., 2025, Lemma 1) Let the scale parameter $\lambda = 1/\epsilon_0$, the range parameter $R = 2\mu$ and the mean parameter is at least $\mu \geq 1 + \frac{1}{\epsilon_0} \log\left(\frac{1}{2\delta_0}\left(1 - e^{-\mu\epsilon_0}\right)\right)$. Then, adding such $(\mu, \lambda, R)$ TBL noise to a 1-sensitivity function provides $(\epsilon_0, \delta_0)$-DP.*

Stated differently, for a given pair $(\mu, \epsilon_0)$, the smallest achievable $\delta_0$ is given by

$$
\delta_0 \geq \tfrac{1}{2} e^{-(\mu-1)\epsilon_0}\left(1 - e^{-\mu\epsilon_0}\right).
\tag{12}
$$

It is evident that for a fixed $\epsilon_0$, larger values of $\mu$ allow smaller values of $\delta_0$ to be chosen. Let $\tilde{Z} \geq 0$ be drawn from a $(\mu, \lambda, R)$ TBL distribution. We propose to privatize $f(\mathcal{G})$ via $\tilde{f}(\mathcal{G}) = f(\mathcal{G}) - \tilde{Z}$. If the parameters of the TBL distribution are chosen according to Fact 1, then $\tilde{f}(\cdot)$ is $(\epsilon_0, \delta_0)$-DP.

• **Analyzing the sensitivity of $\phi(\cdot)$:** After privatizing $f(\mathcal{G})$, we finally consider the optimization problem

$$
\begin{aligned}
\phi(\mathcal{G}) = \min\ & d(\mathcal{G}, \mathcal{G}') \\
\text{s.t.}\ & \theta(\mathcal{G}') \geq \beta \cdot u(\tilde{f}(\mathcal{G})),
\end{aligned}
\tag{13}
$$

In order to perform PTR with $\phi(\cdot)$, we need to compute its $\ell_1$-sensitivity under edge-DP. Let $\mathsf{GS}_\phi$ denote the global sensitivity of $\phi(\cdot)$. In our framework, it turns out that $\mathsf{GS}_\phi$ can be larger than 1, unlike the modified PTR framework of Li et al. (2024). This is showcased by the following result.

**Lemma 2.** *The $\ell_1$ global sensitivity of $\phi(\cdot)$ satisfies*

$$\mathsf{GS}_\phi = \begin{cases} S_1 := 2 + (2 - \sqrt{2})\mu, & -1 < \tilde{f}(G) < 1, \\ S_2 := 1, & otherwise. \end{cases} \tag{14}$$

*Proof.* Please refer to Appendix G. □

After having described the construction of $\phi(\cdot)$ and analyzing its sensitivity, we are ready to establish the privacy properties of the PTR algorithm.

**Theorem 3.** *For every graph $\mathcal{G}$, the PTR algorithm is $(\epsilon_0 + \epsilon_1 + \epsilon_2, \delta_0 + \delta)$ differentially-private, where $\delta_0$ is computed based on equation 12.*

*Proof.* Please refer to Appendix H. □

Taking a step back, its worthwhile to contrast our modified PTR algorithm with that of Li et al. (2024). Although we adopt the same idea as Li et al. (2024), the main difference is that we need to perform an extra differentially private check to test if the given dataset lies in the large graph regime or not. If the latter case arises, the algorithm will stop w.p. $1 - \delta$. The second difference is that the sensitivity of $\phi(\cdot)$ depends on the parameter $\mu$ used in the TBL, which is a consequence of the fact that the proposed bound used in problem equation 9 depends on the outcome of $u(\tilde{f}(\mathcal{G}))$. These modifications necessitate a larger privacy budget compared to the PTR algorithm proposed in Li et al. (2024). Next, we turn to the practical considerations of selecting $\beta$ and evaluating $\phi$.

• **Valid choices of $\beta$:** Selecting the parameter $\beta$ is a key component of implementing PTR. The following theorem specifies the range of $\beta$ values that are valid for the PTR algorithm.

**Theorem 4.** *For any choice of $\beta \in (\beta_l, \beta_u)$, the statistic $\phi(\cdot)$ can be computed according to equation 17, where the lower and upper limits are given by*

$$\beta_l := \frac{2\sqrt{v_{\pi(1)}^2 + v_{\pi(2)}^2}}{GAP(\mathcal{G})}, \beta_u := \frac{2\sqrt{2}}{GAP(\mathcal{G})}\left[2 - \sqrt{2} + \sqrt{v_{\pi(1)}^2 + v_{\pi(2)}^2}\right]. \tag{15}$$

*Proof.* Please refer to Appendix I. □

• **Computing $\phi(\cdot)$:** Given a $\beta \in (\beta_l, \beta_u)$, the tractability of the PTR framework hinges on our ability to solve problem 7 in an efficient manner in order to compute $\phi(\cdot)$. By construction, the problem asks to find the smallest value of $d(\mathcal{G}, \mathcal{G}')$ such that the inequality

$$\frac{2}{\mathsf{GAP}(\mathcal{G}) - d(\mathcal{G}, \mathcal{G}')} \cdot \left[\frac{2d(\mathcal{G}, \mathcal{G}')}{\mathsf{GAP}(\mathcal{G})} + \sqrt{v_{\pi(1)}^2 + v_{\pi(2)}^2}\right] \geq \beta \tag{16}$$

is valid. Observe that for a given value of $\beta$, all the involved parameters can be readily computed from $\mathcal{G}$. Furthermore, the fact that $\theta(\mathcal{G}')$ (i.e., the LHS of the above inequality) is monotonically increasing with $d(\mathcal{G}, \mathcal{G}')$, facilitates simple solution. In fact, for a given $\beta$, the statistic $\phi(\cdot)$ can be computed in *closed form* according to the formula

$$\phi(\mathcal{G}) = \left\lceil \frac{\beta \cdot \mathsf{GAP}^2(\mathcal{G}) - 2\mathsf{GAP}(\mathcal{G})\sqrt{v_{\pi(1)}^2 + v_{\pi(2)}^2}}{4 + \beta \cdot \mathsf{GAP}(\mathcal{G})} \right\rceil. \tag{17}$$

• **Policy for selecting** $\beta$ : Finally, we need to have a policy for selecting a suitable $\beta \in (\beta_l, \beta_u)$. From equation 17, it is evident that $\phi(\mathcal{G})$ increases monotonically with $\beta \in (\beta_l, \beta_u)$. Hence, for a fixed set of privacy parameters $(\epsilon_1, \delta)$, larger values of $\beta$ correspond to higher likelihood of the noisy statistic $\hat{\phi}(\mathcal{G})$ exceeding the threshold $(GS_\phi \ln(1/\delta))/\epsilon_1$, which in turn increases the odds of the PTR algorithm yielding a response. However, larger values of $\beta$ also lead to increased noise injection in the release step of PTR. Hence, a suitable $\beta$ should strike the "sweet spot" between minimizing noise injection and maximizing the odds of a successful response. Next, we demonstrate how the choice of $\beta$ impacts these two conflicting objectives.

For a given $\beta$, the PTR algorithm *succeeds* if the value of the noisy statistic $\hat{\phi}(\mathcal{G})$ exceeds the threshold $(GS_\phi \ln(1/\delta))/\epsilon_1$, which implies that the algorithm responds with an output according to step 5. Hence, the *success probability* of PTR corresponds to the event $\mathsf{Prob}(\hat{\phi}(\mathcal{G}) \geq (GS_\phi \log(1/\delta))/\epsilon_1)$, where the randomness is w.r.t. the Laplace random variable $\mathsf{Lap}(0, 1/\epsilon_1)$. We now reveal how the success probability is affected by the choice of $\beta$. To this end, it will be convenient to express $\phi$ as $\phi(\mathcal{G}) = \lceil \tau(\mathcal{G}) \rceil$ where $\tau(\mathcal{G})$ is the fraction in equation 17. Note that by design, we have $\lfloor \tau(\mathcal{G}) \rfloor + 1 \geq \phi(\mathcal{G}) \geq \tau(\mathcal{G})$. Since $\tau(.)$ increases monotonically with $\beta$, increasing $\tau(.)$ also increases $\phi(\cdot)$, which in turn, directly influences the outcome of the threshold test. Next, we demonstrate that controlling the value of $\tau(.)$ through $\beta$ is sufficient to derive lower bounds on the success probability of PTR.

**Theorem 5.** *Let $\phi(\mathcal{G}) = \lceil \tau(\mathcal{G}) \rceil$ where $\tau(\mathcal{G})$ is the fraction in equation 17. If $\beta$ is selected to satisfy*

$$\tau(\mathcal{G}) = (p + GS_\phi) \cdot \frac{\log(1/\delta)}{\epsilon_1}, \forall \, p \in (0, 1], \tag{18}$$

*then the success probability of PTR is at least $1 - \frac{\delta^p}{2}$.*

*Proof.* Please refer to Appendix J. □

Intuitively, the smaller the value of $p$, the closer the value of $\phi(\cdot)$ is to the threshold $(GS_\phi \log(1/\delta))/\epsilon_1$, and hence the lower the probability of success. This is captured by the above theorem, which shows that as $p$ decreases, the success probability diminishes at the rate $1 - O(\delta^{(p-1)})$. As $p \to 1$, the lower bound on the success probability shrinks to $1/2$.

The value of $\beta$ which satisfies equation 18 is given by

$$\beta = \frac{2}{\mathsf{GAP}(\mathcal{G})} \cdot \left\lceil \frac{2(p + GS_\phi)\log(1/\delta)/\epsilon_1 + \mathsf{GAP}(\mathcal{G})\sqrt{v_{\pi(1)}^2 + v_{\pi(2)}^2}}{\mathsf{GAP}(\mathcal{G}) - (p + GS_\phi)\log(1/\delta)/\epsilon_1} \right\rceil \tag{19}$$

We can obtain the following insights from the above formula. First, fixing all parameters in equation 19 except $p$, we see that $\beta$ diminishes as $p$ is reduced. We conclude that smaller values of $p$ result in the injection of smaller amounts of noise, but this comes at the expense of reduced odds of the algorithm succeeding. Hence, the parameter $p$ directly controls the trade-off between noise-injection levels and the success probability. Second, fixing all parameters except the graph dependent quantities $\mathsf{GAP}(\mathcal{G})$ and $\sqrt{v_{\pi(1)}^2 + v_{\pi(2)}^2}$, we observe that $\beta = O\left(\frac{\sqrt{v_{\pi(1)}^2 + v_{\pi(2)}^2}}{\mathsf{GAP}(\mathcal{G})}\right)$, which implies that graphs with a large spectral gap and small spread in the energy of the entries of $\mathbf{v}$ are amenable to smaller levels of noise injection. In Appendix K, we show that the family of expander graphs Hoory et al. (2006), which mimic several properties of social networks Malliaros & Megalooikonomou (2011), fulfill both conditions.

We conclude with a final sanity check to ensure that $\beta$ computed according to equation 19 lies in the interval $(\beta_l, \beta_u)$.

**Proposition 1.** *For a fixed graph with spectral gap $\mathsf{GAP}(\mathcal{G})$, and parameters $\epsilon_1, \delta, p$, the condition*

$$\frac{\log(1/\delta)}{\epsilon_1} < (1 - 1/\sqrt{2})\frac{\mathsf{GAP}(\mathcal{G})}{(p + GS_\phi)} \implies \beta \in (\beta_l, \beta_u).$$

*Proof.* Please refer to Appendix L. □

---

**Algorithm 1:** PPC via PTR $(\mathbf{A}, p, \epsilon_0, \epsilon_1, \epsilon_2, \delta)$

---

**Input:** Symmetric $A \in \mathbb{R}^{n \times n}$, $p$, $\mathbf{v}$, $\mu$, privacy parameters $\epsilon_0, \epsilon_1, \epsilon_2$, $\delta > 0$.
**Output:** Private PC $\mathbf{v}_{\text{ptr}}$.
Compute $\delta_0$ based on equation 12.
Compute $\beta$ based on equation 19.
`// Phase I: Private gap test:  check whether the graph is well-behaved`
Sample $z \sim \text{TBL}(\mu, \lambda = 1, R = 2\mu)$.
Compute $\tilde{f}(G) = f(G) - z$.
**if** $\tilde{f}(G) \geq 0$ **then**
$\quad|\quad$ Compute $\phi(\mathcal{G})$ based on equation 17.
**else**
$\quad|\quad$ $\phi(\mathcal{G}) = 0$.
**end**
`// Phase II: Private distance computation`
**if** $-1 < \tilde{f}(G) < 1$ **then**
$\quad|\quad$ Set $\mathsf{GS}_\phi = 2 + (2 - \sqrt{2})\mu$.
**else**
$\quad|\quad$ Set $\mathsf{GS}_\phi = 1$.
**end**
Compute $\hat{\phi}(\mathcal{G}) = \phi(\mathcal{G}) + \text{Lap}\left(\frac{\mathsf{GS}_\phi}{\epsilon_1}\right)$.

`// Phase III: Private release via output perturbation`
**if** $\hat{\phi}(\mathcal{G}) \geq \frac{\mathsf{GS}_\phi \ln(1/\delta)}{\epsilon_1}$ **then**
$\quad|\quad$ Compute $\mathbf{v}_{\text{ptr}} = \mathbf{v} + \mathcal{N}\left(0, \frac{2\beta^2 \log(2/\delta)}{\epsilon_2^2} \cdot \mathbf{I}_n\right)$.
$\quad|\quad$ **Return:** $\mathbf{v}_{\text{ptr}} = \frac{\mathbf{v}_{\text{ptr}}}{\|\mathbf{v}_{\text{ptr}}\|_2}$.
**else**
$\quad|\quad$ **Return:** No Response.
**end**

---

The final proposed solution to approximate D$k$S via PTR is presented in Algorithm 1. Note that the steps of the algorithm are in closed-form, and hence it can be executed extremely quickly.

## 6   The Private Power Method

We consider the Private Power Method (PPM) of Hardt & Price (2014) as a baseline for computing private principal components under edge-DP. In essence, PPM is a "noisy" variant of the classic (non-private) power method which computes the principal component of a matrix in an iterative fashion. Executing each step entails performing a matrix-vector multiplication, followed by normalization. In Hardt & Price (2014), this algorithm is made DP by adding Gaussian noise to each matrix-vector multiplication, followed by normalization. By calibrating the variance of the Gaussian noise added in each iteration to the $\ell_2$-sensitivity of matrix-vector multiplication, the final iterate of PPM (after a pre-determined number of iterations have been carried out) can be shown to satisfy $(\epsilon, \delta)$-DP. Hence, in contrast to PTR, PPM is not a one-shot noise addition scheme. Furthermore, its output set is a singleton - a noisy principal component satisfying $(\epsilon, \delta)$-DP, which can reflect a smaller privacy budget compared to PTR. The privacy model considered in Hardt & Price (2014) defines a pair of matrices $\mathbf{A}$ and $\mathbf{A}'$ to be neighboring if they differ in one entry by at-most 1. As per the authors of Hardt & Price (2014), this is most meaningful when the entries of the data matrix $\mathbf{A}$ lie in $[0, 1]$. Note that when $\mathbf{A}$ and $\mathbf{A}'$ correspond to a pair of neighboring adjacency matrices, this naturally coincides with the notion of edge-DP for undirected graphs.

The complete algorithm is described in Algorithm 2. According to (Hardt & Price, 2014, Theorem 1.3), after $L = O\left(\frac{\lambda_1 \log n}{\mathsf{GAP}(\mathcal{G})}\right)$ iterations, the output $\mathbf{v}_L$ of PPM satisfies $(\epsilon, \delta)$-DP and with probability at least $9/10$, it holds that

$$\|(\mathbf{I} - \mathbf{v}_L \mathbf{v}_L^T)\mathbf{v}\|_2 \leq O\left(\frac{\sigma \cdot \max_{\ell \in [L]} \|\mathbf{v}_L\|_\infty \cdot \sqrt{n \log L}}{\mathsf{GAP}(\mathcal{G})}\right). \tag{20}$$

In terms of computational complexity, each step of PPM requires computing a sparse matrix-vector multiplication, which incurs $O(m)$ complexity, followed by noise addition and re-normalization, which incurs an additional $O(n)$ time. Hence, each step has complexity order $O(n + m)$. After $L$ iterations, the overall complexity is $O((n + m)L)$. If $L$ is chosen according to (Hardt & Price, 2014, Theorem 1.3), this results in complexity of order $O((\lambda_1(n + m) \log n)/\mathsf{GAP}(\mathcal{G}))$.

---

**Algorithm 2:** PPC via PPM $(\mathbf{A}, L, \epsilon, \delta)$

---

**Input:** Symmetric $A \in \mathbb{R}^{n \times n}$, no. of iterations $L$, privacy parameters $\epsilon$, $\delta > 0$.
**Output:** Private PC $\mathbf{v}_{\mathsf{ppm}}$.

**Initialize:** Let $\mathbf{v}_0$ be a random unit direction and set $\sigma = \epsilon^{-1}\sqrt{4L \log(1/\delta)}$.
**for** $\ell = 1$ **to** $L$ **do**
 Generate $\mathbf{g}_\ell \sim \mathcal{N}(0, \|\mathbf{v}_{\ell-1}\|_\infty^2 \sigma^2) \in \mathbb{R}^n$
 Compute $\mathbf{w}_\ell = A\mathbf{v}_{\ell-1} + \mathbf{g}_\ell$
 Normalize $\mathbf{v}_\ell = \mathbf{w}_\ell/\|\mathbf{w}_\ell\|_2$
**end**
**Return:** $\mathbf{v}_{\mathsf{ppm}} = \mathbf{v}_L$

---

## 7 Applications

Let $\mathbf{v}_{\mathsf{priv}}$ denote a $(\epsilon, \delta)$ edge-DP estimate of $\mathbf{v}$. In this section, we explain how $\mathbf{v}_{\mathsf{priv}}$ can be utilized for the following applications.

• **(A1) Private top-$k$ eigenscore subset extraction:** In the non-private scenario, computing the subset with the $k$ largest eigenscores is equivalent to solving the problem

$$\mathbf{x}_k = \arg \max_{\mathbf{x} \in \mathcal{X}_k} \mathbf{v}^T \mathbf{x}, \tag{21}$$

since $\mathbf{v}$ is element-wise non-negative. The solution is given by the support of the $k$-largest entries of $\mathbf{v}$, and can be accomplished in $O(n \log k)$ time. In the private setting, $\mathbf{v}_{\mathsf{priv}}$ need not be element-wise non-negative in general. Hence, in this case, we solve the problem

$$\hat{\mathbf{x}}_k = \arg \max_{\mathbf{x} \in \mathcal{X}_k} |\mathbf{v}_{\mathsf{priv}}^T \mathbf{x}| \tag{22}$$

Note that the post-processing property of DP implies that $\hat{\mathbf{x}}_k$ also obeys $(\epsilon, \delta)$-DP. In order to solve the above problem, define the pair of candidate solutions

$$\mathbf{x}_k^{(1)} = \arg \max_{\mathbf{x} \in \mathcal{X}_k} \mathbf{v}_{\mathsf{priv}}^T \mathbf{x}; \quad \mathbf{x}_k^{(2)} = \arg \min_{\mathbf{x} \in \mathcal{X}_k} \mathbf{v}_{\mathsf{priv}}^T \mathbf{x} \tag{23}$$

Note that $\mathbf{x}_k^{(1)}$ and $\mathbf{x}_k^{(2)}$ correspond to the support of the $k$-largest and $k$-smallest entries of $\mathbf{v}_{\mathsf{priv}}$ respectively. Between these two candidates, the one which attains the larger objective value corresponds to the solution of equation 22; i.e., we have

$$\hat{\mathbf{x}}_k = \arg \max_{i \in \{1,2\}} |\mathbf{v}_{\mathsf{priv}}^T \mathbf{x}_k^{(i)}|. \tag{24}$$

Thus, the post-processing step remains computationally efficient, as it only requires examining an additional candidate solution compared to its non-private counterpart 21.

• **(A2) Private D$k$S:** Problem equation 2 is NP-hard and difficult to approximate Manurangsi (2017); Jones et al. (2023). As a result, we resort to the low-rank approximation scheme of Papailiopoulos et al. (2014), which uses the principal component $\mathbf{v}$ to approximate D$k$S. The best rank-one approximation of the adjacency matrix $\mathbf{A}$ is denoted as $\hat{\mathbf{A}} := \lambda_1 \mathbf{v}\mathbf{v}^T$. Applying the rank-one approximation of $\mathbf{A}$ to the objective function of problem equation 2 is equivalent to solving the problem $\mathbf{x}_k = \max_{\mathbf{x} \in \mathcal{X}_k} \mathbf{v}^T \mathbf{x}$, which is the same as problem 21, and be solved in $O(n \log k)$ time. The work of Papailiopoulos et al. (2014) showed that such a subset of nodes constitute dense subgraphs in real-datasets, and provide a data-dependent approximation guarantee for D$k$S (see Appendix M).

Table 2: Summary of network statistics: the number of vertices ($n$), the number of edges ($m$), the eigen-gap, and the network type.

| Graph | $n$ | $m$ | Eigen-gap | Network Type |
|-------|-----|-----|-----------|--------------|
| FACEBOOK | 4K | 88K | 36.9 | Social |
| PPI-HUMAN | 21K | 342K | 70.8 | Biological |
| SOC-BLOGCATALOG | 89K | 2.1M | 335.9 | Social |
| FLICKR | 106K | 2.32M | 101.3 | Image relationship |
| TWITCH-GAMERS | 168K | 6.8M | 148.4 | Social |
| ORKUT | 3.07M | 117.19M | 225.4 | Social |

In the private setting, we now solve the following rank-1 approximation problem

$$\hat{\mathbf{x}}_k = \arg \max_{\mathbf{x} \in \mathcal{X}_k} |\mathbf{v}_{\mathsf{priv}}^T \mathbf{x}| \tag{25}$$

to obtain the final output $\hat{\mathbf{x}}_k$. Since $\mathbf{v}_{\mathsf{priv}}$ may not be element-wise non-negative, as argued previously, it suffices to examine the top-$k$ and bottom-$k$ support of $\mathbf{v}_{\mathsf{priv}}$, and then output the candidate that achieves a larger objective value.

## 8 Experimental Results

Here, we test the efficacy of the proposed DP algorithms on real graph datasets in terms of their privacy-utility trade-off for applications **(A1)** and **(A2)** and their runtime.

### 8.1 Setup

• **Datasets:** Table 2 provides a summary of the datasets, which were sourced from standard repositories Kunegis (2013); Jure (2014). All experiments were performed in Python on a Linux work-station with 132GB RAM and an Intel i7 processor. We used the following settings.

(1) **PTR:** For the TBLM, we set $(\epsilon_0, \delta_0) = (1, 7e^{-7})$ and $\mu = 3t \approx 14.48$. We also set the other privacy budget parameters as ($\epsilon_1 = \epsilon_2 = \epsilon = 3$), while we set $\delta = \log(m)/m$, where $m$ is the number of edges in each graph. We set the lower bound on the success probability of obtaining a response in Algorithm 1 to be 0.95, and then, based on Theorem 5, set $p = 1 - (1/\log_{10}(\delta))$.

**Handling No responses:** A potential issue with PTR mechanisms is the frequency of returning NO RESPONSE. In our framework, this can happen due to (a): the dataset not clearing the private gap test in phase I or (b): the dataset clearing the gap test but failing the threshold test for release in Phase III. For the latter case, Theorem 5 provides an explicit lower bound on the success probability, which allows end-users to directly control the probability of successful private release via specification of the parameter $p$. In all experiments, the datasets we used successfully cleared the gap test, and we selected $p$ so that the probability of release is at least 0.95. We empirically observed that PTR returns a valid output in the vast majority of trial runs across all datasets (see Table 4). For general datasets, if a "no response" is obtained, one could repeat the PTR mechanism (since it is very fast) or increase the lower bound on the success probability to improve the odds of a response. However, this also increases $\beta$ and results in addition of greater levels of noise to the principal component. In the event that the mechanism repeatedly returns "no response", then one could fall back to using PPM.

(2) **PPM:** We set ($\epsilon = 3$) and the parameter $\delta$ was fixed to be ($\delta = \log(m)/m$) across all datasets. An open question with practical implementation of PPM is how to select the number of iterations $L$, which has to be specified beforehand. The main issue is that (Hardt & Price, 2014, Theorem 1.3) only provides the order of iterations required to achieve a certain utility bound, which scales like $O(\frac{\lambda_1 \log n}{\mathsf{GAP}(\mathcal{G})})$. We experimentally observed that simply setting $L = \frac{\lambda_1 \log n}{\mathsf{GAP}(\mathcal{G})}$ provides good empirical performance across the different datasets.

Table 3: Execution time to privatize the PC.

| Graph | PPM (ms) | PTR (ms) | Speedup |
|---|---|---|---|
| FACEBOOK | 13.6 | 0.07 | 194 |
| PPI-HUMAN | 58.6 | 0.32 | 183 |
| SOC-BLOGCATALOG | 283 | 1.64 | 172 |
| FLICKR | 587 | 1.83 | 320 |
| TWITCH-GAMERS | 7781 | 2.25 | 3458 |
| ORKUT | 29660 | 43.14 | 688 |

Performing more iterations typically degraded the performance of the algorithm, in addition to also increasing the run-time complexity.

Further results for a more detailed investigation of the proposed solutions across different values of the privacy parameters can be found in Appendix N.

| FACEBOOK | PPI-HUMAN | SOC-BLOGCATALOG | FLICKR | TWITCH-GAMERS | ORKUT |
|---|---|---|---|---|---|
| 98% | 99.6% | 98.6% | 96.9% | 99.6% | 98.5% |

Table 4: Empirical success rates of the PTR algorithm across different datasets with $(\epsilon_0, \delta_0) = (1, 7 \times 10^{-7}), \epsilon_1 = \epsilon_2 = 3, \delta = \log(m)/m$, $(m)$: the number of edges.

## 8.2 Results

• **Timing:** As shown in Table 3, PTR significantly outperforms PPM in terms of execution time required to output a private PC. We obtain at least a 170-fold speedup across all datasets, and on the TWITCH-GAMERS dataset we obtain a 3500-fold speedup. This is because the steps of the PTR algorithm can be computed in closed form, and it adds noise once, which makes it fast. On the other hand, PPM incurs $O(n + m)$ complexity with every iteration, which makes the overall algorithm slower. These results underscore the practical benefits of our PTR algorithm. Given that the original PTR algorithm Dwork & Lei (2009) is not known to be polynomial-time, our modifications and insightful parameter selection come together to result in a fast algorithm.

• **(A1) Private Top-$k$ eigenscore subset extraction:** In order to measure utility, for each private subset extracted via Algorithms 1 and 2, we compute its Jaccard similarity with the non-private subset. The results for the 4 datasets with the largest eigen-gaps are depicted in Figure 3. Each sub-figure plots the Jaccard similarity versus subset size $k$ for both private algorithms on a specific dataset (averaged across 200 Monte-Carlo trials). For these datasets, the two algorithms offer comparable utility under a modest privacy budget - in fact, the private top-$k$ subsets exhibit high similarity with their non-private counterpart. PTR incurs a larger overall privacy budget compared to PPM (roughly twice more in terms of the $\epsilon$ parameter) to attain the same utility. This can be attributed to the fact that PTR outputs 3 noisy parameters privately, whereas PPM only outputs only one. As explained before, from a utility-time complexity perspective, PTR performs better in general.

• **(A2) Private D$k$S:** The empirical performance of Algorithms 1 and 2 across different datasets is depicted in Figure 4. Each sub-figure plots the edge-density versus size curve obtained using the two different algorithms. As a baseline, we consider the non-private algorithm as described in Papailiopoulos et al. (2014). In addition, we also include an upper bound for the edge density of the D$k$S (See appendix M). To generate each plot, we executed Algorithm 1 and and Algorithm 2 for 100 different realizations, with each private PC used to generate a different edge density-size curve. For each subgraph size $k$, we depict the average edge density attained by each method across all realizations, within one standard deviation (vertical lines). From Figure 4, it is evident that both PTR and PPM output subgraphs whose edge-density closely matches that of

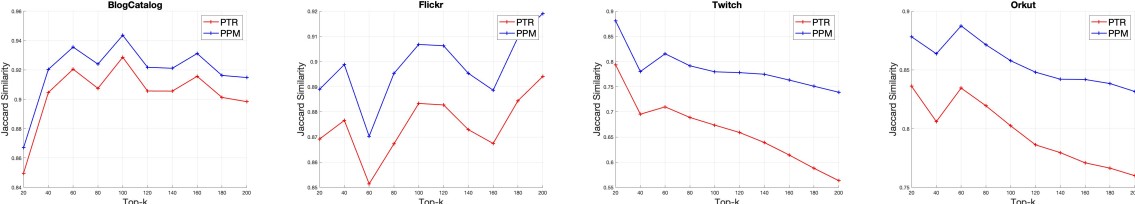

Figure 3: Results for private top-$k$ eigenscore subset detection. Jaccard similarity (y-axis) vs. subset size $k$ (x-axis): PTR (red) $(\epsilon_0, \delta_0) = (1, 7 \times 10^{-7}), \epsilon_1 = \epsilon_2 = 3, \delta_1 = \log(m)/m$, PPM (blue) $\epsilon = 3, \delta = \log(m)/m$, non-private (yellow), ($m$): the number of edges.

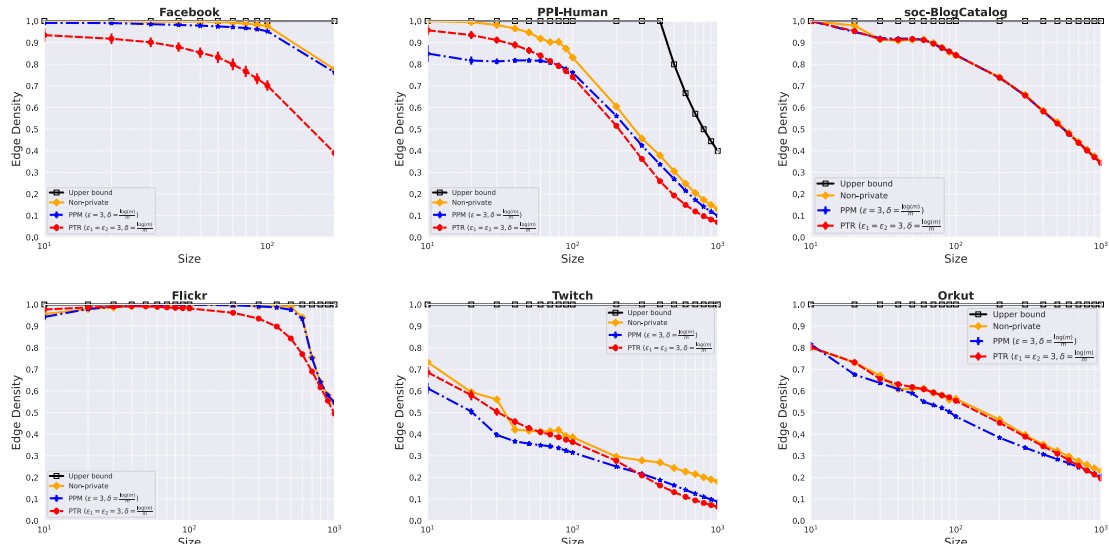

Figure 4: Results for private D$k$S. Edge density (y-axis) vs. subgraph size $k$ (x-axis): PTR (red) $(\epsilon_0, \delta_0) = (1, 7 \times 10^{-7})$, PPM (blue), non-private (yellow), and the upper bound on the maximum attainable edge density (black). ($m$): the number of edges. Higher densities are better.

the non-private solution, again for a modest privacy budget of $\epsilon \geq 3$. Again, PTR is faster in extracting highly-quality private dense subsets, at the expense of a larger privacy budget.

Based on our findings, we conclude that both PTR and PPM offer good utility in the applications considered compared to the non-private baselines. The advantage of PTR is its transparent parameter selection and its computational efficiency, allowing it to scale to large networks much more easily, at the cost of a slightly higher privacy budget.

# 9 Conclusions

In this paper, we considered the problem of privately computing the principal component of the graph adjacency matrix under edge-DP. Motivated by the large gap between the local and global sensitivity on real-world datasets, we employed the Propose-Test-Release (PTR) framework. Owing to its instance specific nature, PTR can offer good utility on well-behaved datasets by injecting small amounts of noise to provide DP. However, it is computationally expensive. To overcome this challenge, we develop a new practical and powerful PTR framework which obviates the computational complexity issues inherent in the standard framework, while facilitating simple selection of the algorithm parameters. As a consequence, our PTR algorithm also results in the first DP algorithm for the densest-$k$-subgraph (D$k$S) problem, a key graph mining primitive. We test our approach on real-world graphs, and demonstrate that it can attain performance

comparable to the non-private solution while adhering to a modest privacy budget. Compared to an iterative baseline based on the private power method (PPM), PTR requires a slightly larger privacy budget, but is more than two orders of magnitude faster on average. DP techniques need to be scalable to larger datasets, for privacy practices and guarantees to become more commonplace. Our paper makes advances towards this goal.

## 10  Acknowledgments

Alireza Khayatian and Aritra Konar were supported by the KU Leuven Special Research Fund BOF/STG-22-040. Anil Vullikanti was partially supported by NSF grants CCF-1918656 and CNS-2317193, DTRA award HDTRA1-24-R-0028, and the Commonwealth Cyber Initiative Cybersecurity Research Award.

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

# A  Supporting Lemmata

## A.1  The Davis-Kahan-sin$\Theta$ Theorem

Consider the $n \times n$ symmetric matrix $\mathbf{M}^*$ with eigen-decomposition $\mathbf{M}^* = \sum_{i=1}^{n} \lambda_i^* \mathbf{u}_i^* \mathbf{u}_i^{*T}$, where $|\lambda_1^*| \geq |\lambda_2^*| \geq \cdots \geq |\lambda_n^*|$ denote the eigen-values of $\mathbf{M}^*$ sorted in descending order and $\{\mathbf{u}_i^*\}_{i \in [n]}$ are the corresponding eigen-vectors. Let $\mathbf{M}$ be a $n \times n$ symmetric matrix obtained by perturbing $\mathbf{M}$; i.e., we have

$$\mathbf{M} = \mathbf{M}^* + \mathbf{E}. \tag{26}$$

In an analogous manner, the eigen-decomposition of $\mathbf{M}$ is defined as $\mathbf{M} = \sum_{i=1}^{n} \lambda_i \mathbf{u}_i \mathbf{u}_i^T$. Let $\mathbf{U} = [\mathbf{u}_1, \cdots, \mathbf{u}_r]$ and $\mathbf{U}^* = [\mathbf{u}_1^*, \cdots, \mathbf{u}_r^*]$ denote the principal-$r$ eigen-spaces associated with $\mathbf{M}$ and $\mathbf{M}^*$ respectively. The distance between the principal eigen-spaces $\mathbf{U}$ and $\mathbf{U}^*$ can be defined as

$$\mathsf{dist}(\mathbf{U}, \mathbf{U}^*) := \min_{\mathbf{Q} \in \mathcal{O}^{r \times r}} \|\mathbf{UQ} - \mathbf{U}^*\|_2,$$

where $\mathcal{O}^{r \times r}$ denotes the set of $r \times r$ rotation matrices. The classic result of Davis-Kahan Davis & Kahan (1970) provides eigen-space perturbation bounds in terms of the strength of the perturbation $\mathbf{E}$ and the eigen-gap of $\mathbf{M}^*$. While there are many variations of this result Stewart (1990), we utilize one variant which will prove particularly useful in our context (Chen et al., 2021, Corollary 2.8).

**Theorem 6.** *If the perturbation satisfies $\|\mathbf{E}\|_2 \leq (1 - 1/\sqrt{2})(|\lambda_r^*| - |\lambda_{r+1}^*|)$, then the distance between the principal eigen-spaces $\mathbf{U}$ and $\mathbf{U}^*$ obeys*

$$\mathit{dist}(\mathbf{U}, \mathbf{U}^*) \leq \frac{2\|\mathbf{EU}^*\|_2}{|\lambda_r^*| - |\lambda_{r+1}^*|}. \tag{27}$$

In particular, when $r = 1$, we obtain the following bound on the distance between the principal components of $\mathbf{M}$ and $\mathbf{M}^*$ as a corollary.

**Corollary 1.** *Under the conditions of Theorem 6, the distance between the principal components $\mathbf{u}_1$ and $\mathbf{u}_1^*$ is bounded by*

$$\mathit{dist}(\mathbf{u}_1, \mathbf{u}_1^*) \leq \frac{2\|\mathbf{Eu}_1^*\|_2}{|\lambda_1^*| - |\lambda_2^*|} \tag{28}$$

## B    Proof of Theorem 1

Consider a pair of neighboring graphs $\mathcal{G}, \mathcal{G}'$ and let $\mathbf{A}, \mathbf{A}'$ denote their adjacency matrices respectively. Under edge-DP, we can view $\mathbf{A}'$ as a perturbation of $\mathbf{A}$; i.e., we have

$$\mathbf{A}' = \mathbf{A} + \mathbf{E}, \tag{29}$$

where $\mathbf{E}$ models the affect of adding/deleting an edge in $\mathcal{G}$. Such an action can be formally expressed as

$$\mathbf{E} = \begin{cases} \mathbf{e}_i\mathbf{e}_j^T + \mathbf{e}_j\mathbf{e}_i^T, & i \neq j, (i,j) \notin \mathcal{E} \\ -(\mathbf{e}_i\mathbf{e}_j^T + \mathbf{e}_j\mathbf{e}_i^T), & i \neq j, (i,j) \in \mathcal{E} \end{cases} \tag{30}$$

where $\mathbf{e}_i$ denotes the $i^{th}$ canonical basis vector. When $(i,j) \notin \mathcal{E}$, the perturbation $\mathbf{E}$ models an edge addition to $\mathcal{G}$, whereas for $(i,j) \in \mathcal{E}$, $\mathbf{E}$ models an edge deletion. Note that by construction, $\mathbf{E}$ is a sparse matrix with a pair of symmetric non-zero entries $E_{ij} = E_{ji} = \pm 1$, and satisfies $\|\mathbf{E}\|_2 = 1$. We denote the set of all such possible perturbations $\mathbf{E}$ obtained by edge addition/removal as $\mathcal{P}$.

Let $\mathbf{v}$ and $\mathbf{v}'$ denote the principal components of $\mathbf{A}$ and $\mathbf{A}'$ respectively. Since $\mathbf{A}$ and $\mathbf{A}'$ have non-negative entries, the Perron-Frobenius theorem implies that $\mathbf{v}$ and $\mathbf{v}'$ are also element-wise non-negative. Hence, the distance between them can be expressed as

$$\mathsf{dist}(\mathbf{v}, \mathbf{v}') = \min_{\alpha \in \{-1,+1\}} \|\mathbf{v} - \alpha\mathbf{v}'\|_2 = \|\mathbf{v} - \mathbf{v}'\|_2 \tag{31}$$

We will invoke the Davis-Kahan perturbation bound stated in Corollary 1 in order to upper bound the local $\ell_2$ sensitivity of $\mathbf{v}$. This is valid provided the perturbation $\mathbf{E}$ defined in equation 30 satisfies

$$\|\mathbf{E}\|_2 \leq (1 - 1/\sqrt{2})(|\lambda_1^*| - |\lambda_2^*|)$$
$$\Leftrightarrow |\lambda_1^*| - |\lambda_2^*| \geq \frac{\|\mathbf{E}\|_2}{1 - 1/\sqrt{2}} \tag{32}$$
$$\Leftrightarrow \mathsf{GAP}(\mathcal{G}) \geq \frac{1}{1 - 1/\sqrt{2}} = \frac{\sqrt{2}}{\sqrt{2} - 1}$$

Hence, for graphs whose eigen-gap exceeds $\sqrt{2}/(\sqrt{2}-1)$, we can estimate the local sensitivity via the following chain of inequalities.

$$\mathsf{LS}_\mathbf{v}(\mathcal{G}) = \max_{\mathcal{G}':\mathcal{G}\sim\mathcal{G}'} \|\mathbf{v} - \mathbf{v}'\|_2$$
$$\leq \max_{\mathbf{E}\in\mathcal{P}} \frac{2\|\mathbf{E}\mathbf{v}\|_2}{|\lambda_1| - |\lambda_2|}$$
$$= \max_{\substack{i\in\mathcal{V},j\in\mathcal{V}, \\ i\neq j}} \left\{ \frac{2\sqrt{v_i^2 + v_j^2}}{\mathsf{GAP}(\mathcal{G})} \right\} \tag{33}$$
$$\leq \frac{2c_\pi}{\mathsf{GAP}(\mathcal{G})}$$

where in the final step, $v_{\pi(1)}$ and $v_{\pi(2)}$ are the largest and second-largest elements of $\mathbf{v}$, respectively. The first inequality follows from Corollary 1, the second equality is a consequence of the structured sparsity exhibited by $\mathbf{E}$, and the final inequality follows since $\sqrt{v_{\pi(1)}^2 + v_{\pi(2)}^2} \geq \sqrt{v_i^2 + v_j^2}, \forall\, i \in \mathcal{V}, \forall\, j \in \mathcal{V}, i \neq j$.

## C    Proof of Theorem 2

Let $\mathbf{v}'$ and $\mathbf{v}''$ denote the principal components of the adjacency matrices $\mathbf{A}'$ and $\mathbf{A}''$ associated with $\mathcal{G}'$ and a neighboring dataset $\mathcal{G}''$, respectively; i,e, we have $\mathbf{A}'' = \mathbf{A}' + \mathbf{E}'$, where $\mathbf{E}'$ models the addition/removal of an edge. Applying the Davis-Kahan-$\sin\Theta$ Theorem, we obtain

$$\mathsf{LS}(\mathcal{G}') = \max_{\mathcal{G}''\sim\mathcal{G}'} \|\mathbf{v}' - \mathbf{v}''\|_2 \leq \frac{2\max_{\mathbf{E}'\in\mathcal{P}} \|\mathbf{E}'\mathbf{v}'\|_2}{\mathsf{GAP}(\mathcal{G}')}. \tag{34}$$

We have implicitly made the assumption that $\mathsf{GAP}(\mathcal{G}') > \frac{\|\mathbf{E}'\|_2}{1 - 1/\sqrt{2}} = \frac{1}{1 - 1/\sqrt{2}}$, since $\|\mathbf{E}'\|_2 = 1$. Later, we will show that this assumption is satisfied under conditions (A1) and (A2).

To establish the desired result, we individually upper and lower bound the numerator and denominator terms of the above bound on $\mathsf{LS}(\mathcal{G}')$. First, consider the numerator term. Then, we have

$$
\begin{aligned}
\max_{\mathbf{E}' \in \mathcal{P}} \|\mathbf{E}'\mathbf{v}'\|_2 &= \max_{\mathbf{E}' \in \mathcal{P}} \|\mathbf{E}'(\mathbf{v}' - \mathbf{v} + \mathbf{v})\|_2 \\
&\leq \max_{\mathbf{E}' \in \mathcal{P}} \|\mathbf{E}'(\mathbf{v}' - \mathbf{v})\|_2 + \max_{\mathbf{E}' \in \mathcal{P}} \|\mathbf{E}'\mathbf{v}\|_2 \\
&\leq \|\mathbf{v}' - \mathbf{v}\|_2 \cdot \max_{\mathbf{E}' \in \mathcal{P}} \|\mathbf{E}'\|_2 + \max_{\mathbf{E}' \in \mathcal{E}} \|\mathbf{E}'\mathbf{v}\|_2 \\
&\leq \|\mathbf{v}' - \mathbf{v}\|_2 + \sqrt{v_{\pi(1)}^2 + v_{\pi(2)}^2} \\
&= \mathsf{dist}(\mathbf{v}', \mathbf{v}) + \sqrt{v_{\pi(1)}^2 + v_{\pi(2)}^2}
\end{aligned}
\tag{35}
$$

Viewing $\mathbf{A}'$ as a perturbation of $\mathbf{A}$, we can apply the $\sin\Theta$ theorem of Davis-Kahan to bound $\mathsf{dist}(\mathbf{v}', \mathbf{v})$ in terms of $\mathbf{v}$ and $\mathsf{GAP}(\mathcal{G})$. Let $\mathbf{E} = \mathbf{A}' - \mathbf{A}$ denote the perturbation that transforms $\mathbf{A}$ to $\mathbf{A}'$. Note that we have $\|\mathbf{E}\|_2 \leq d(\mathcal{G}, \mathcal{G}')$. Applying the $\sin\Theta$ theorem then yields

$$
\mathsf{dist}(\mathbf{v}', \mathbf{v}) \leq \frac{2\|\mathbf{E}\mathbf{v}\|_2}{\mathsf{GAP}(\mathcal{G})} \leq \frac{2\|\mathbf{E}\|_2\|\mathbf{v}\|_2}{\mathsf{GAP}(\mathcal{G})} \leq \frac{2d(\mathcal{G}, \mathcal{G}')}{\mathsf{GAP}(\mathcal{G})},
\tag{36}
$$

provided $\|\mathbf{E}\|_2 < (1 - 1/\sqrt{2}) \cdot \mathsf{GAP}(\mathcal{G})$. Since $\|\mathbf{E}\|_2 \leq d(\mathcal{G}, \mathcal{G}')$, this condition is satisfied if $d(\mathcal{G}, \mathcal{G}') < (1 - 1/\sqrt{2}) \cdot \mathsf{GAP}(\mathcal{G})$, which is assumption (A1). Combining equation 35 and equation 36, we obtain

$$
\max_{\mathbf{E}' \in \mathcal{P}} \|\mathbf{E}'\mathbf{v}'\|_2 \leq \frac{2d(\mathcal{G}, \mathcal{G}')}{\mathsf{GAP}(\mathcal{G})} + \sqrt{v_{\pi(1)}^2 + v_{\pi(2)}^2},
\tag{37}
$$

provided the condition listed in assumption (a1) holds.

Next, consider the denominator term of equation 34. To obtain a lower bound, the following lemma will prove useful.

**Lemma 3.** *(Gonem & Gilad-Bachrach, 2018, Lemma 11) Let $\mathcal{G}$ and $\mathcal{G}'$ be a pair of graphs with $d(\mathcal{G}, \mathcal{G}') = k$, and suppose $GAP(\mathcal{G}) > 0$. Then*

$$
\max\{GAP(\mathcal{G}) - k, 0\} \leq GAP(\mathcal{G}') \leq GAP(\mathcal{G}) + k.
\tag{38}
$$

If we wish to apply the non-trivial version of the lower bound on $\mathsf{GAP}(\mathcal{G}')$, then we require $d(\mathcal{G}, \mathcal{G}') < \mathsf{GAP}(\mathcal{G})$. Note that under assumption (A1), this condition is already satisfied. Hence, we obtain

$$
\mathsf{GAP}(\mathcal{G}') \geq \mathsf{GAP}(\mathcal{G}) - d(\mathcal{G}, \mathcal{G}').
\tag{39}
$$

On combining equation 37 and equation 39, we obtain the upper bound

$$
\frac{2\max_{\mathbf{E}' \in \mathcal{P}} \|\mathbf{E}'\mathbf{v}'\|_2}{\mathsf{GAP}(\mathcal{G}')} \leq \frac{2}{\mathsf{GAP}(\mathcal{G}) - d(\mathcal{G}, \mathcal{G}')} \cdot \left[ \frac{2d(\mathcal{G}, \mathcal{G}')}{\mathsf{GAP}(\mathcal{G})} + \sqrt{v_{\pi(1)}^2 + v_{\pi(2)}^2} \right]
\tag{40}
$$

It only remains to chain the above inequality with equation 34. However, the inequalities equation 34 and equation 40 were derived under different assumptions, and a little care must be taken to ensure that they hold simultaneously. Note that equation 34 requires that $\mathsf{GAP}(\mathcal{G}') > \sqrt{2}/(\sqrt{2} - 1)$. If assumption (A1) is satisfied, we know that the lower bound equation 39 applies. Hence, if $\mathsf{GAP}(\mathcal{G}) - d(\mathcal{G}, \mathcal{G}') > \sqrt{2}/(\sqrt{2} - 1)$, it implies that $\mathsf{GAP}(\mathcal{G}') > \sqrt{2}/(\sqrt{2} - 1)$. It remains to work out what is the minimum value of $\mathsf{GAP}(\mathcal{G})$ required so that (A1) and $\mathsf{GAP}(\mathcal{G}) - d(\mathcal{G}, \mathcal{G}') > \sqrt{2}/(\sqrt{2} - 1)$ are both valid. Under (A1), we have

$$
\mathsf{GAP}(\mathcal{G}) - d(\mathcal{G}, \mathcal{G}') > \mathsf{GAP}(\mathcal{G}).(1/\sqrt{2})
\tag{41}
$$

If $\mathsf{GAP}(\mathcal{G}).(1/\sqrt{2}) > \sqrt{2}/(\sqrt{2} - 1)$, it implies the desired condition $\mathsf{GAP}(\mathcal{G}) - d(\mathcal{G}, \mathcal{G}') > \sqrt{2}/(\sqrt{2} - 1)$. The former condition is satisfied by $\mathsf{GAP}(\mathcal{G}) > 2/(\sqrt{2} - 1)$, which is assumption (A2).

To conclude, under assumptions (A1) and (A2), we are free to chain together the inequalities equation 34 and equation 40. Doing so yields the claimed bound

$$\mathsf{LS}(\mathcal{G}') \leq \frac{2}{\mathsf{GAP}(\mathcal{G}) - d(\mathcal{G}, \mathcal{G}')} \cdot \left[\frac{2d(\mathcal{G}, \mathcal{G}')}{\mathsf{GAP}(\mathcal{G})} + \sqrt{v_{\pi(1)}^2 + v_{\pi(2)}^2}\right].$$

This concludes the proof.

## D  The difficulty with smooth sensitivity

Smooth sensitivity Nissim et al. (2007) is a framework for obtaining DP guarantees while relying on local sensitivity based quantities to calibrate the level of injected noise, as opposed to using the global sensitivity. To be specific, the *smooth sensitivity*, for a graph dataset $\mathcal{G}$ is defined as

$$S_f^\beta(\mathcal{G}) := \max_{\mathcal{G}'} \left\{\mathsf{LS}_f(\mathcal{G}') \cdot \exp(-\beta d(\mathcal{G}, \mathcal{G}'))\right\}. \tag{42}$$

Here, $f : \mathcal{G} \to \mathbb{R}^n$ denotes the target function to be privatized, $\beta > 0$ is a parameter, and $d(\mathcal{G}, \mathcal{G}')$ denotes the Hamming distance between $\mathcal{G}$ and $\mathcal{G}'$. By adding i.i.d. Gaussian noise to $f$ that is calibrated to $S_f^\beta(\mathcal{G})$, it can be shown that the resulting output satisfies DP (with $\beta$ reflecting the desired privacy parameters $(\epsilon, \delta)$). The smooth sensitivity value $S_f^\beta(\mathcal{G})$ can be viewed as the tightest upper bound on the local sensitivity that provides DP. However, the catch is that solving the optimization problem equation 42 is non-trivial in general, which renders practical application of the smooth sensitivity framework challenging. The prior work of Gonem & Gilad-Bachrach (2018) employed the smooth sensitivity framework for computing principal components of general datasets; albeit not for graphs under edge-DP. Using the local sensitivity estimate in (Gonem & Gilad-Bachrach, 2018, Theorem 5), the authors developed tractable smooth upper bounds on $S_f^\beta(\mathcal{G})$, which can then be used to provide DP. However, successfully adapting this approach to our present context presents several difficulties. In D, we provide a rigorous analysis which shows that under mild conditions, the obtained smooth upper bound is very close to the global sensitivity estimate of $\sqrt{2}$.

In this section, we illustrate the difficulties associated with adopting the smooth sensitivity framework of Gonem & Gilad-Bachrach (2018); Nissim et al. (2007) for computing principal components of $\mathcal{G}$ under edge DP.

Following Nissim et al. (2007), for a graph $\mathcal{G}$, the $\beta$-smooth sensitivity of the principal component $\mathbf{v}$ with smoothness parameter $\beta > 0$ can be defined as

$$S_{\mathbf{v}}^\beta(\mathcal{G}) = \max_{t \in \{0, 1, \cdots, \binom{n}{2}\}} \left\{e^{-\beta t} \cdot \gamma_{\mathbf{v}}^{(t)}(\mathcal{G})\right\}, \tag{43}$$

where

$$\gamma_{\mathbf{v}}^{(t)}(\mathcal{G}) := \max_{\mathcal{G}':d(\mathcal{G},\mathcal{G}')=t} \mathsf{LS}_{\mathbf{v}}(\mathcal{G}') \tag{44}$$

is the local $\ell_2$-sensitivity of $\mathbf{v}$ at Hamming distance $t$ from $\mathcal{G}$.

Since computing $S_{\mathbf{v}}^\beta(\mathcal{G})$ exactly can prove to be challenging, we adopt the approach of Gonem & Gilad-Bachrach (2018) to obtain smooth upper bounds on its value. To this end, the following result of (Nissim et al., 2007, Claim 3.2) is useful.

**Lemma 4.** *For an admissible $t_0 \in \{0, 1, \cdots, \binom{n}{2}\}$, let*

$$\hat{S}_{\mathbf{v}}^\beta(\mathcal{G}) := \max\left(\max_{t \in \{0, \cdots, t_0-1\}} \{e^{-\beta t} \cdot \gamma_{\mathbf{v}}^{(t)}(\mathcal{G})\}, \ \mathsf{GS}_{\mathbf{v}} \cdot e^{-\beta t_0}\right), \tag{45}$$

*where $\mathsf{GS}_{\mathbf{v}}$ denotes the global $\ell_2$-sensitivity of $\mathbf{v}$. Then, $\hat{S}_{\mathbf{v}}^\beta(\mathcal{G})$ is a $\beta$-smooth upper bound.*

In order to compute the above smooth upper bound, we will utilize the local $\ell_2$-sensitivity estimate of $\mathbf{v}$ derived in Theorem 1, which is valid for all graphs with a spectral gap of at least $\nu := \sqrt{2}/(\sqrt{2}-1)$. The

main idea is now to modify the proof technique of Gonem & Gilad-Bachrach (2018) so that this condition can be incorporated. To this end, define the following "gap-restricted" analogue of equation 44.

$$\hat{\gamma}_{\mathbf{v}}^{(t)}(\mathcal{G}) := \max_{\substack{\mathcal{G}':d(\mathcal{G},\mathcal{G}')=t,\\ \mathsf{GAP}(\mathcal{G}')>\nu}} \mathsf{LS}_{\mathbf{v}}(\mathcal{G}'). \tag{46}$$

Note that in general, we have $\hat{\gamma}^{(t)}(\mathcal{G}) \leq \gamma^{(t)}(\mathcal{G})$. However, from the lower bound in Lemma 2, we know that for $t < \mathsf{GAP}(\mathcal{G}) - \nu$, we have $\mathsf{GAP}(\mathcal{G}') > \nu$. We conclude that

$$\gamma_{\mathbf{v}}^{(t)}(\mathcal{G}) = \hat{\gamma}_{\mathbf{v}}^{(t)}(\mathcal{G}), \forall\, t < \mathsf{GAP}(\mathcal{G}) - \nu \tag{47}$$

Next, we invoke Theorem 2, which asserts that

$$\mathsf{LS}(\mathcal{G}') \leq \frac{2}{\mathsf{GAP}(\mathcal{G}) - t} \cdot \left[ \frac{2t}{\mathsf{GAP}(\mathcal{G})} + \sqrt{v_{\pi(1)}^2 + v_{\pi(2)}^2} \right] \tag{48}$$

subject to the conditions $t < (1 - 1/\sqrt{2}) \cdot \mathsf{GAP}(\mathcal{G}) := \mathsf{GAP}(\mathcal{G})/\nu$ and $\mathsf{GAP}(\mathcal{G}) > \frac{2}{\sqrt{2}-1} := \sqrt{2}\nu$. Furthermore, a little calculation reveals that the assumption

$$\mathsf{GAP}(\mathcal{G}) > \sqrt{2}\nu \implies \frac{\mathsf{GAP}(\mathcal{G})}{\nu} < \mathsf{GAP}(\mathcal{G}) - \nu, \tag{49}$$

which will prove useful. In particular, from equation 47 and equation 49, we obtain that for graphs with spectral gap at least $\sqrt{2}\nu$

$$\gamma_{\mathbf{v}}^{(t)}(\mathcal{G}) = \hat{\gamma}_{\mathbf{v}}^{(t)}(\mathcal{G}), \forall\, t \leq \frac{\mathsf{GAP}(\mathcal{G})}{\nu}. \tag{50}$$

We are now free to use equation 48 to upper bound $\gamma_{\mathbf{v}}^{(t)}(\mathcal{G})$. Doing so yields

$$\begin{aligned} \gamma_{\mathbf{v}}^{(t)}(\mathcal{G}) &\leq \frac{2}{\mathsf{GAP}(\mathcal{G}) - t} \cdot \left[ \frac{2t}{\mathsf{GAP}(\mathcal{G})} + \sqrt{v_{\pi(1)}^2 + v_{\pi(2)}^2} \right], \\ &\leq \frac{2}{\mathsf{GAP}(\mathcal{G}) - t} \cdot \left[ \frac{2}{\nu} + \sqrt{u_{\pi(1)}^2 + u_{\pi(2)}^2} \right], \\ &\leq \frac{2\sqrt{2}}{\mathsf{GAP}(\mathcal{G})} \left[ \frac{2}{\nu} + \sqrt{v_{\pi(1)}^2 + v_{\pi(2)}^2} \right], \forall\, t < \frac{\mathsf{GAP}(\mathcal{G})}{\nu}. \end{aligned} \tag{51}$$

We are now ready to apply Lemma 4. Let $t_0 = \frac{\mathsf{GAP}(\mathcal{G})}{\nu}$. Then, we have

$$e^{-\beta t} \cdot \gamma_{\mathbf{v}}^{(t)}(\mathcal{G}) \leq \gamma_{\mathbf{v}}^{(t)}(\mathcal{G}) \leq \frac{2\sqrt{2}}{\mathsf{GAP}(\mathcal{G})} \left[ \frac{2}{\nu} + \sqrt{v_{\pi(1)}^2 + v_{\pi(2)}^2} \right], \forall\, t < t_0 \tag{52}$$

This yields the following smooth sensitivity bound for the principal component of graphs with eigen-gap larger than $2\nu$.

$$\hat{S}_{\mathbf{v}}^{\beta}(\mathcal{G}) = \max\left\{ \frac{2\sqrt{2}}{\mathsf{GAP}(\mathcal{G})} \left[ \frac{2}{\nu} + \sqrt{v_{\pi(1)}^2 + v_{\pi(2)}^2} \right], \sqrt{2}e^{-\beta \frac{\mathsf{GAP}(\mathcal{G})}{\nu}} \right\} \tag{53}$$

Let us examine the obtained smooth upper bound. In order to apply this bound to obtain approximate $(\epsilon, \delta)$-DP via the Gaussian mechanism, (Nissim et al., 2007, Lemma 2.7) asserts that

$$\beta = \frac{\epsilon}{4(n + \ln(2/\delta))}. \tag{54}$$

This reveals the primary drawback - namely the dependence of $\beta$ on the dimension $n$. Since $\beta = O(\epsilon/n)$ (discounting $\delta$ for the moment), for even moderately sized graphs on $n$ vertices, the value of $\beta$ can be very

small (i.e., $\ll 1$) for reasonable choices of privacy parameters $(\epsilon, \delta)$. We conclude that the smooth upper bound

$$
\begin{aligned}
\hat{S}_\mathbf{v}^\beta(\mathcal{G}) &= \sqrt{2}\exp\left(-O\left(\frac{\epsilon\mathsf{GAP}(\mathcal{G})}{n}\right)\right) \\
&\approx \sqrt{2}\left(1 - O\left(\frac{\epsilon\mathsf{GAP}(\mathcal{G})}{n}\right)\right)
\end{aligned}
\tag{55}
$$

where the approximation in the last step holds when $n \gg \epsilon\mathsf{GAP}(\mathcal{G})$. For the datasets considered in this paper, we observed that computing the smooth upper bound with the same privacy budget allotted to PTR yields values close to $\sqrt{2}$, which is the global sensitivity value.

## E    Proof of Lemma 1

The function $\theta(\mathcal{G}')$ is monotonically increasing in $d(\mathcal{G}, \mathcal{G}')$ for all feasible datasets $\mathcal{G}'$ for which $\theta(\mathcal{G}') \geq 0$. Hence, the minimum is attained by selecting $\mathcal{G} = \mathcal{G}'$, from which it follows that $\psi(\mathcal{G}) = 0$.

## F    Sampling noise from the TBL distribution

Let $F$ be the CDF of the *untruncated* Laplace$(\mu, \lambda)$:

$$
F(x) = \begin{cases} \dfrac{1}{2}\,e^{(x-\mu)/\lambda}, & x \leq \mu, \\[2mm] 1 - \dfrac{1}{2}\,e^{-(x-\mu)/\lambda}, & x > \mu. \end{cases}
$$

Truncating to $[0, R]$ means conditioning on that interval. The truncated CDF $G$ is

$$
G(x) = \Pr[X \leq x \mid 0 \leq X \leq R] = \frac{F(x) - F(0)}{F(R) - F(0)} \quad (x \in [0, R]).
$$

Hence the *inverse* truncated CDF is

$$
G^{-1}(u) = F^{-1}\big(F(0) + u\,[F(R) - F(0)]\big) \quad (u \in (0, 1)).
$$

Sampling algorithm:

1. Draw $u \sim \mathrm{Unif}(0, 1)$, set $u' = F(0) + u\,[F(R) - F(0)]$.

2. Invert the *untruncated* Laplace CDF at $u'$:

$$
X = \begin{cases} \mu_L + \lambda_L \ln\big(2u'\big), & u' \leq \frac{1}{2}, \\[2mm] \mu_L - \lambda_L \ln\big(2(1 - u')\big), & u' > \frac{1}{2}. \end{cases}
$$

Note the branch test must be against $u'$ (the *untruncated* CDF value), not $u$, unless the truncation is symmetric ($R = 2\mu$), in which special case the branch also happens at $u = 1/2$.

**Closed forms for $F(0)$ and $F(R)$.** For $\mu > 0$ and $R \geq \mu$,

$$
F(0) = \tfrac{1}{2}e^{-\mu/\lambda}, \qquad F(R) = 1 - \tfrac{1}{2}e^{-(R-\mu)/\lambda},
$$

so the truncated interval mass is $Z_{\mu,\lambda,R} = F(R) - F(0) = 1 - \tfrac{1}{2}e^{-(R-\mu)/\lambda} - \tfrac{1}{2}e^{-\mu/\lambda}$, and $R = 2\mu$ we get $Z_{\mu,\lambda,R} = 1 - e^{-\mu/\lambda}$.

# G   Proof of Lemma 2

Contrary to the standard PTR and modified PTR Li et al. (2024), here $\mathsf{GS}_\phi$ is not necessarily equal to 1.

To compute the global sensitivity of $\phi(G)$, it is essential to account for the step preceding this part of the algorithm. In the previous stage, we obtained the private function $\tilde{f}(G)$, whose value directly influences the computation of $\phi(G)$. Since $\tilde{f}(G)$ is produced in an earlier step, we can treat it as a constant when analyzing the global sensitivity of $\phi(G)$. In other words, the value of $\tilde{f}(G)$ is regarded as prior knowledge in the sensitivity analysis of $\phi(G)$.

Let $\mathcal{G} \sim \mathcal{G}''$ be a pair of neighboring instances. Then, for a fixed draw of $\tilde{Z} = z$, we consider the following two cases.

**Case 1:** $u(\tilde{f}(\mathcal{G})) = u(\tilde{f}(\mathcal{G}''))$. The value equals either 0 or 1. In the case of the former, from Lemma 1, we obtain $\phi(\mathcal{G}) = \phi(\mathcal{G}'') = 0$, and the sensitivity is 0. Otherwise, the sensitivity is 1 by a standard argument.

**Case 2:** $u(\tilde{f}(\mathcal{G})) \neq u(\tilde{f}(\mathcal{G}''))$. Suppose that $u(\tilde{f}(\mathcal{G})) = 0$ but $u(\tilde{f}(\mathcal{G}'')) = 1$. Then, we have $\phi(\mathcal{G}) = 0$ and $\phi(\mathcal{G}'') \geq 0$. Hence, the sensitivity of $\phi$ is $\phi(\mathcal{G}'')$. In order to determine how large this quantity can be, we use the following facts: (1) $\tilde{f}(\mathcal{G}) \leq 0$ and $\tilde{f}(\mathcal{G}'') > 0$, and (2) $|\mathsf{GAP}(\mathcal{G}) - \mathsf{GAP}(\mathcal{G}'')| \leq 1$. From the first fact, we obtain $\mathsf{GAP}(\mathcal{G}) \leq t + z$, whereas the second fact together with $\mathsf{GAP}(\mathcal{G}'') > t + z$ yield the boundary condition $\mathsf{GAP}(\mathcal{G}) > t + z - 1$. Hence, this condition arises when the gap of $\mathcal{G}$ lies in the interval $(t + z - 1, t + z]$, which is equivalent to $\tilde{f}(\mathcal{G}) \in (-1, 0]$. Since $\mathsf{GAP}(\mathcal{G}'') > t + z > t$, $\mathcal{G}''$ lies in the large gap regime and thus from [Theorem 2, (A2)], it holds that

$$\phi(\mathcal{G}'') < \left(1 - \frac{1}{\sqrt{2}}\right) \cdot \mathsf{GAP}(\mathcal{G}'') \leq \left(1 - \frac{1}{\sqrt{2}}\right) \cdot (\mathsf{GAP}(\mathcal{G}) + 1) \leq \left(1 - \frac{1}{\sqrt{2}}\right) \cdot (t + z + 1) \leq \left(1 - \frac{1}{\sqrt{2}}\right) \cdot (t + 2\mu + 1) \tag{56}$$

where in the final inequality, we have used the fact that $z \in [0, 2\mu]$.

Now consider the opposite case when $u(\tilde{f}(\mathcal{G})) = 1$ but $u(\tilde{f}(\mathcal{G}'')) = 0$. Then, $\phi(\mathcal{G}) \geq 0$ and $\phi(\mathcal{G}'') = 0$. Hence, the sensitivity is determined by how large $\phi(\mathcal{G})$ can be. To upper-bound this quantity, we proceed as before. Since $\tilde{f}(\mathcal{G}) > 0$, we obtain $\mathsf{GAP}(\mathcal{G}) > t + z$. Meanwhile, from $\tilde{f}(\mathcal{G}'') \leq 0$ and the fact that the gap has sensitivity 1, we obtain the boundary condition $\mathsf{GAP}(\mathcal{G}) \leq t + z + 1$. Hence, this scenario arises when the gap of $\mathcal{G}$ lies in the interval $(t + z, t + z + 1]$, which is equivalent to $\tilde{f}(\mathcal{G}) \in (0, 1]$. Since $\mathcal{G}$ lies in the large gap regime, from [Theorem 2, (A2)], we obtain

$$\phi(\mathcal{G}) < \left(1 - \frac{1}{\sqrt{2}}\right) \cdot \mathsf{GAP}(\mathcal{G}) \leq \left(1 - \frac{1}{\sqrt{2}}\right) \cdot (t + z + 1) \leq \left(1 - \frac{1}{\sqrt{2}}\right) \cdot (t + 2\mu + 1) \tag{57}$$

All in all, to determine the sensitivity of $\phi(G)$, we first check whether $-1 \leq \tilde{f}(G) \leq 1$. If this condition holds, then $\mathsf{GS}_\phi = (1 - \frac{1}{\sqrt{2}})(t + 2\mu + 1)$. Otherwise, $\mathsf{GS}_\phi = 1$.

# H   Proof of Theorem 3

The key step is to establish that the computation of $\hat{\phi}(\mathcal{G})$ is private for all for graphs. Note that $\hat{\phi}(\mathcal{G})$ can be viewed as the output of an adaptive composition mechanism. The first mechanism is the TBLM for privatizing $f(\mathcal{G})$, whose output $\tilde{f}(\mathcal{G})$ is $(\epsilon_0, \delta_0)$-DP, as established by Fact 1. Furthermore, $\tilde{f}(\mathcal{G})$ is then applied as an input to problem 13, whose solution $\phi(\mathcal{G})$ is then privatized via the standard Laplace mechanism. In order to invoke adaptive composition, we need to show that conditioned on the input $\tilde{f}(\mathcal{G})$, $\hat{\phi}(\mathcal{G})$ is DP. To this end, note that if one replaces the private output of TBLM $\tilde{f}(\mathcal{G})$ with its non-private counterpart $f(\mathcal{G})$ in Lemma 2, then the scale parameters $S_1, S_2$ in equation 14 correspond to the local sensitivity of $\phi(\cdot)$. Clearly, adding Laplacian noise scaled to these parameters is not guaranteed to preserve edge-DP. However, if we select the scale parameter $\mathsf{GS}_\phi$ on the basis of $\tilde{f}(\mathcal{G})$, as in equation 13, then we have that

$$\mathsf{GS}_\phi = S_1 \cdot \mathbf{1}_{\{-1 < \tilde{f}(\mathcal{G}) < 1\}} + S_2 \cdot \mathbf{1}_{\{\tilde{f}(\mathcal{G}) \leq -1\} \cup \{\tilde{f}(\mathcal{G}) \geq 1\}} \tag{58}$$

By the post-processing property of DP, it follows that $\mathsf{GS}_\phi$ is also $(\epsilon_0, \delta_0)$ edge-DP. Hence, using $\mathsf{GS}_\phi$ in the Laplace mechanism to privatize $\phi(\cdot)$ is guaranteed to be $(\epsilon_0, 0)$-DP. Direct application of adaptive composition

then guarantees that the total privacy budget needed to privatize $\phi(\cdot)$ is $(\epsilon_0 + \epsilon_1, \delta_0)$. We are now ready to state the privacy of the overall algorithm.

Depending on the value of $\tilde{f}(\mathcal{G})$, the following outcomes are possible.

**Case 1:** $\tilde{f}(\mathcal{G}) \leq 0$ : The proposed bound in problem 13 is 0, and $\phi(\mathcal{G}) = 0$. From Lemma 2, $\hat{\phi}(\mathcal{G}) \sim$ $\mathsf{Lap}(GS_\phi/\epsilon_1)$. By properties of the Laplace distribution, the probability that $\hat{\phi}(\mathcal{G}) \leq (GS_\phi \ln(1/\delta))/\epsilon_1$ in the test stage of PTR is at least $1 - \delta$, and the algorithm refuses to yield a response for such a dataset. Otherwise, with probability at most $\delta$, the algorithm is not private. We conclude that the output of the test stage of the algorithm is $(0, \delta)$-DP. From basic composition, the overall privacy offered by the algorithm totals $(\epsilon_0 + \epsilon_1, \delta_0 + \delta)$.

**Case 2:** $\tilde{f}(\mathcal{G}) > 0$ : The proposed bound in problem 13 is $\beta$. The remainder of the analysis is broken down into two further two sub-cases which depend on $\beta$. First, let $\beta < \mathsf{LS}_\mathbf{v}(\mathcal{G})$. Then $\gamma(\mathcal{G}) = 0$, which implies that $\phi(\mathcal{G}) = 0$, since $\gamma(\mathcal{G}) \geq \phi(\mathcal{G}) \geq 0$. By the same argument as the previous case, the probability that $\hat{\phi}(\mathcal{G}) \leq (GS_\phi \ln(1/\delta))/\epsilon_1$ and the algorithm stops is at least $1 - \delta$. Hence the overall privacy privacy totals $(\epsilon_0 + \epsilon_1, \delta_0 + \delta)$. In the other sub-case, $\beta \geq \mathsf{LS}_\mathbf{v}(\mathcal{G})$, and the overall output is $(\epsilon_0 + \epsilon_1 + \epsilon_2, \delta_0 + \delta)$ DP, being the composition of a and a $(\epsilon_2, \delta)$-DP Gaussian mechanism.

## I Proof of Theorem 4

Selecting the parameter $\beta$ is a key component of implementing PTR. First, we specify an interval of values of $\beta$ which will be considered. We start from verifying the following condition; namely, whether for a given value of $\beta$, the solution of problem equation 16 satisfies $\phi(\mathcal{G}) < (1 - 1/\sqrt{2}) \cdot \mathsf{GAP}(\mathcal{G})$, as this assumption is required in 2 to establish that $\theta(\mathcal{G}')$ is a valid upper bound on $\mathsf{LS}(\mathcal{G}')$. As we show next, ensuring that this condition is met translates into a maximum allowable value of $\beta$.

Since $\theta(\mathcal{G})$ is monotonically increasing with $d(\mathcal{G}, \mathcal{G}')$, the largest value of $\beta$ which can be satisfied occurs when $d(\mathcal{G}, \mathcal{G}') = (1 - 1/\sqrt{2}) \cdot \mathsf{GAP}(\mathcal{G}) - 1$. From equation 16, this corresponds to the following upper bound on $\beta$.

$$\beta_u := \frac{2\sqrt{2}}{\mathsf{GAP}(\mathcal{G})} \left[ 2 - \sqrt{2} + \sqrt{v_{\pi(1)}^2 + v_{\pi(2)}^2} \right] \tag{59}$$

By the same principle, the largest value of $\beta$ for which $d(\mathcal{G}, \mathcal{G}') = 0$ in problem equation 16 occurs for

$$\beta_l := \frac{2\sqrt{v_{\pi(1)}^2 + v_{\pi(2)}^2}}{\mathsf{GAP}(\mathcal{G})}, \tag{60}$$

which corresponds to the upper bound on the local $\ell_2$ sensitivity of $\mathbf{v}$ on $\mathcal{G}$. Hence, for any choice of $\beta \in (\beta_l, \beta_u)$, the statistic $\phi(\cdot)$ can be computed according to equation 17.

## J Proof of Theorem 5

We will utilize the following fact regarding the Laplace distribution. **Fact 1:** Let $Z \sim \mathsf{Lap}(0, b)$. Then,

$$\mathsf{Prob}(|Z| \geq tb) = \exp(-t). \tag{61}$$

In particular, if $b = GS_\phi/\epsilon$ and $t = c \cdot \log(1/\delta)$ (where $c > 0$), then

$$\mathsf{Prob}(|Z| \geq c \cdot \log(1/\delta)/\epsilon) = \delta^c.$$

In the PTR algorithm, after computing $\phi(\cdot)$ in step 2, we add noise $Z \sim \mathsf{Lap}(0, GS_\phi/\epsilon_1)$ to obtain the noisy statistic $\hat{\phi}(\mathcal{G}) = \phi(\mathcal{G}) + Z$. Thereafter, we test whether $\hat{\phi}$ exceeds the threshold $GS_\phi \log(1/\delta)/\epsilon_1$ to yield a response. Since $\phi(\mathcal{G}) \geq \tau(\mathcal{G})$, the probability of a successful response is at least

$$\begin{aligned} \mathsf{Prob}(\hat{\phi}(\mathcal{G}) \geq GS_\phi \log(1/\delta)/\epsilon_1) &= \mathsf{Prob}(Z \geq GS_\phi \log(1/\delta)/\epsilon_1 - \phi(\mathcal{G})) \\ &\geq \mathsf{Prob}(Z \geq GS_\phi \log(1/\delta)/\epsilon_1 - \tau(\mathcal{G})). \end{aligned} \tag{62}$$

Suppose we adjust $\beta$ so that

$$\tau(\mathcal{G}) = (p + GS_\phi) \cdot \log(1/\delta)/\epsilon, \forall\, p \in (0, 1]. \tag{63}$$

Then, the success probability of obtaining a response is at least

$$
\begin{aligned}
\mathsf{Prob}(\hat{\phi}(\mathcal{G}) \geq GS_\phi \log(1/\delta)/\epsilon_1) &\geq \mathsf{Prob}(Z \geq GS_\phi \log(1/\delta)/\epsilon_1 - \tau(\mathcal{G})) \\
&= \mathsf{Prob}(Z \geq p \log(1/\delta)/\epsilon_1) \\
&= 1 - \mathsf{Prob}(Z \leq -p \log(1/\delta)/\epsilon_1) \\
&= 1 - \frac{\exp(-p \log(1/\delta))}{2} \\
&= 1 - \frac{\delta^p}{2},
\end{aligned} \tag{64}
$$

where in the second-last step we have invoked Fact 1 and utilized the fact that the distribution of $Z$ is symmetric about the origin.

## K  Expander graphs

From equation equation 19, it can be seen that for a fixed privacy budget and other algorithm parameters, the proposed bound $\beta$ behaves like

$$\beta \approx O\left(\frac{\sqrt{v_{\pi(1)}^2 + v_{\pi(2)}^2}}{\mathsf{GAP}(\mathcal{G})}\right),$$

where $v_{\pi(1)}, v_{\pi(2)}$ are the two largest entries of the eigen-vector $\mathbf{v}$. This is in line with the local sensitivity bound derived in Theorem 1. Hence, $\beta$ being small depends on (1) the gap being large, *and* (2) the "energy spread" in the entries of $\mathbf{v}$ being small.

We now show that there exists a family of graphs for which both conditions are fulfilled - specifically, the class of expander graphs Hoory et al. (2006). Following the terminology of Hoory et al. (2006), we designate a graph $\mathcal{G}$ as being an $(n, d, \alpha)$-expander if it fulfills the following conditions.

C1: $\mathcal{G}$ has $n$ vertices.

C2: Every vertex has degree $d$, i.e., $\mathcal{G}$ is $d$-regular.

C3: The second largest eigen-value of the adjacency matrix (in magnitude) is $|\lambda_2| \leq \alpha d$, where $\alpha \in (0, 1)$, and $1 - \alpha$ represents the expansion coefficient. Hence, smaller values of $\alpha$ correspond to higher expansion.

For such a family of graphs, the following facts are known.

F1: The principal eigen-vector $\mathbf{v} = \frac{1}{\sqrt{n}}\mathbf{1}_n$.

F2: The largest eigen-value of the adjacency matrix of $\mathcal{G}$ is $\lambda_1 = d$.

F3: For fixed $d$, as $n \to \infty$ it holds that $|\lambda_2| \geq 2\sqrt{d-1} - o_n(1)$

From these facts, we obtain that (a): $v_{\pi(1)}^2 = v_{\pi(2)}^2 = 1/n$, i.e., the entries of $\mathbf{v}$ are uniformly spread out in terms of energy. In addition, (b): the spectral gap $\mathsf{GAP}(\mathcal{G}) = \lambda_1 - |\lambda_2|$ satisfies

$$d(1 - \alpha) \leq \mathsf{GAP}(\mathcal{G}) \leq d - 2\sqrt{d-1} + o_n(1)$$

Hence, the main figure of merit $\frac{\sqrt{v_{\pi(1)}^2 + v_{\pi(2)}^2}}{\mathsf{GAP}(\mathcal{G})}$ can be sandwiched as

$$\frac{\sqrt{2}}{\sqrt{n}[2\sqrt{d-1} + o_n(1)]} \leq \frac{\sqrt{v_{\pi(1)}^2 + v_{\pi(2)}^2}}{\mathsf{GAP}(\mathcal{G})} \leq \frac{\sqrt{2}}{\sqrt{n}d(1 - \alpha)} \tag{65}$$

Suppose that $d(1-\alpha)$ exceeds the threshold $t$ - a quick calculation reveals that $d \geq 12$ is sufficient for every admissible $\alpha \in (0,1)$. As a consequence, $\mathsf{GAP}(\mathcal{G}) \geq t$. Then, for a large $n$ and a fixed $d$, $\beta = \Theta(1/\sqrt{n})$ and the algorithm will release outputs with a small level of noise. Although real-world graphs do not conform precisely to such a model, empirical studies reveal that they can possess good expansion properties Malliaros & Megalooikonomou (2011), which makes them a good candidate for our PTR algorithm.

## L   Proof of Proposition 1

Recall the value of $\beta$ employed in equation 19

$$\beta = \frac{2}{\mathsf{GAP}(\mathcal{G})} \cdot \left[ \frac{2(p + GS_\phi) \cdot \log(1/\delta)/\epsilon_1 + \mathsf{GAP}(\mathcal{G})\sqrt{v_{\pi(1)}^2 + v_{\pi(2)}^2}}{\mathsf{GAP}(\mathcal{G}) - (p + GS_\phi) \cdot \log(1/\delta)/\epsilon_1} \right]. \tag{66}$$

In order to reduce the burden of notation, we refer to the following terms in short-hand.

$$\begin{aligned} a :&= \mathsf{GAP}(\mathcal{G}) \\ b :&= \sqrt{v_{\pi(1)}^2 + v_{\pi(2)}^2} \\ \eta :&= (1 + GS_\phi/p)\log(1/\delta)/\epsilon_1 \end{aligned} \tag{67}$$

Then, $\beta, \beta_l, \beta_u$ can be compactly expressed as

$$\begin{aligned} \beta &= \frac{2}{a} \cdot \left[ \frac{2p\eta + ab}{a - p\eta} \right], \\ \beta_l &= \frac{2b}{a}, \\ \beta_u &= \frac{2\sqrt{2}}{a} \cdot (2 - \sqrt{2} + b), \end{aligned} \tag{68}$$

respectively.

In order to verify for what choices of problem parameters $a, b, \eta, p$ the value of $\beta$ lies in the interval $(\beta_l, \beta_u)$, we consider two cases.

• **Case 1:** $\beta > \beta_l$. This condition is equivalent to

$$\begin{aligned} &\frac{2}{a} \cdot \left[ \frac{2p\eta + ab}{a - p\eta} \right] > \frac{2b}{a} \\ \Leftrightarrow\;& 2p\eta + ab > b(a - p\eta) \\ \Leftrightarrow\;& (2 + b)p\eta > 0, \end{aligned} \tag{69}$$

which is always satisfied provided

$$a - p\eta > 0 \Leftrightarrow \eta < \frac{a}{p}. \tag{70}$$

Note that this corresponds to the necessary condition

$$(p + GS_\phi)\log(1/\delta)/\epsilon_1 < \mathsf{GAP}(\mathcal{G}) \tag{71}$$

for $\beta$ being positive.

- **Case 2:** $\beta < \beta_u$. This condition is equivalent to

$$
\begin{aligned}
&\frac{2}{a} \cdot \left[ \frac{2p\eta + ab}{a - p\eta} \right] < \frac{2\sqrt{2}}{a} \cdot (2 - \sqrt{2} + b) \\
\Leftrightarrow &\frac{2p\eta + ab}{a - p\eta} < \sqrt{2}(2 - \sqrt{2} + b) \\
\Leftrightarrow &2p\eta + ab < (2(\sqrt{2} - 1) + \sqrt{2}b)(a - p\eta) \\
\Leftrightarrow &\eta p(2 + 2(\sqrt{2} - 1) + \sqrt{2}b) < (2(\sqrt{2} - 1) + (\sqrt{2} - 1)b)a \\
\Leftrightarrow &\eta < \frac{(2(\sqrt{2} - 1) + (\sqrt{2} - 1)b)a}{p(2\sqrt{2} + \sqrt{2}b)} \\
\Leftrightarrow &\eta < \frac{(\sqrt{2} - 1)(2 + b)a}{p\sqrt{2}(2 + b)} \\
\Leftrightarrow &\eta < \left( 1 - \frac{1}{\sqrt{2}} \right) \frac{a}{p},
\end{aligned}
\tag{72}
$$

which is the condition

$$
\frac{\log(1/\delta)}{\epsilon_1} < \left( 1 - \frac{1}{\sqrt{2}} \right) \frac{\mathsf{GAP}(\mathcal{G})}{(p + GS_\phi)} < \left( 1 - \frac{1}{\sqrt{2}} \right) \mathsf{GAP}(\mathcal{G}),
\tag{73}
$$

which exactly corresponds to the assumption (A2) in Theorem 2. This completes the proof.

## M  Upper bound of Non-private solution

In terms of the quality of the approximate solution, the following result is known.

**Proposition 2** (Adapted from Papailiopoulos et al. (2014))**.** *For any unweighted graph $\mathcal{G}$, the optimal size-$k$ edge density is no more than*

$$
d_k^* \leq \min \left\{ \frac{1}{k(k-1)} \hat{\mathbf{x}}_k^\top \hat{\mathbf{A}} \hat{\mathbf{x}}_k + \frac{1}{k-1} |\lambda_2|, \frac{1}{k-1} |\lambda_1|, 1 \right\}.
\tag{74}
$$

## N  Additional Experimental Results

In this section, additional results for each of the proposed algorithms are provided individually. The performance of algorithm 1 for D$k$S and top-$k$ eigenscore subset extraction is evaluated in Figure 5 and Figure 7 on real-world datasets for different privacy budgets. Similarly, the performance of algorithm 2 is depicted in Figure 6 and Figure 8.

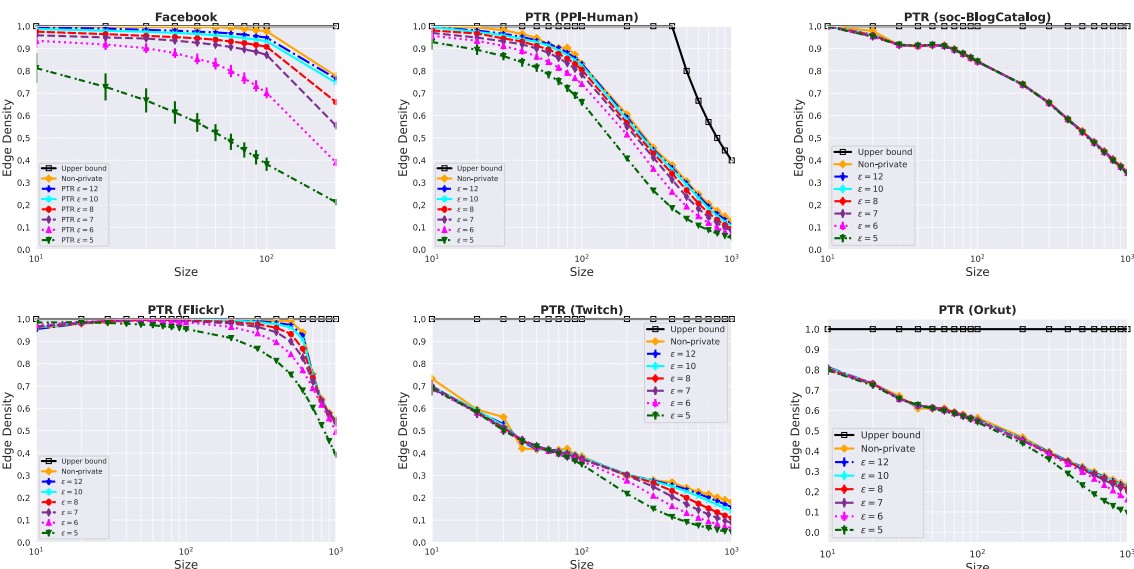

Figure 5: Edge density versus subgraph size ($k$) for Algorithm 1 under $(\epsilon_0, \delta_0) = (1, 7 \times 10^{-7})$ and varying $\epsilon_1 = \epsilon_2 = \epsilon/2$ values across real-world datasets. The privacy parameter is set to $\delta = \log(m)/m$, where $m$ denotes the number of edges in each network.

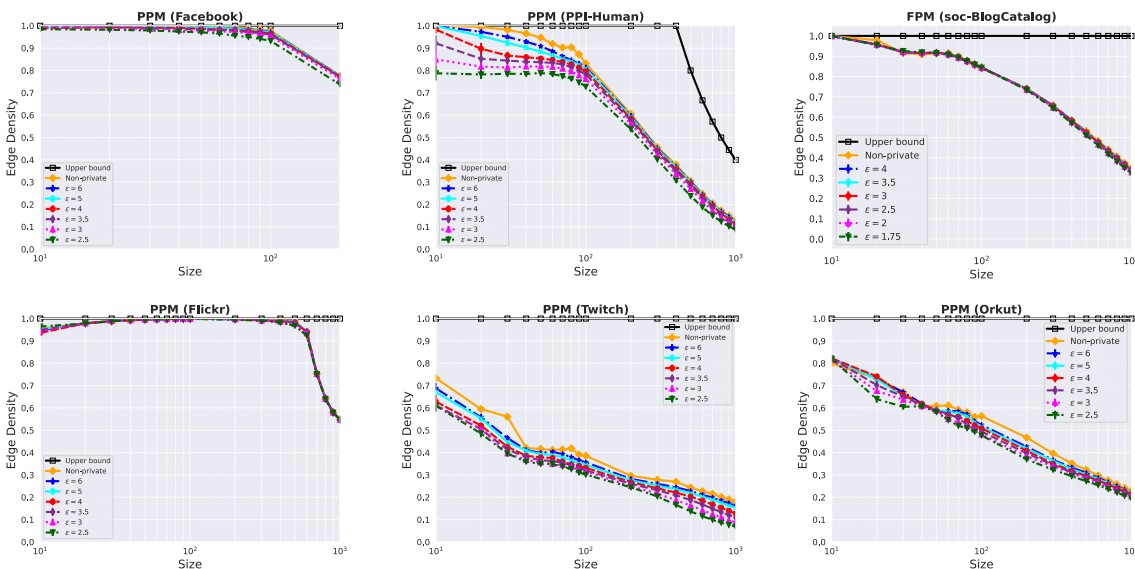

Figure 6: Edge density versus subgraph size ($k$) for Algorithm 2 across real-world datasets under varying $\epsilon$ values with $\delta = \log(m)/m$.

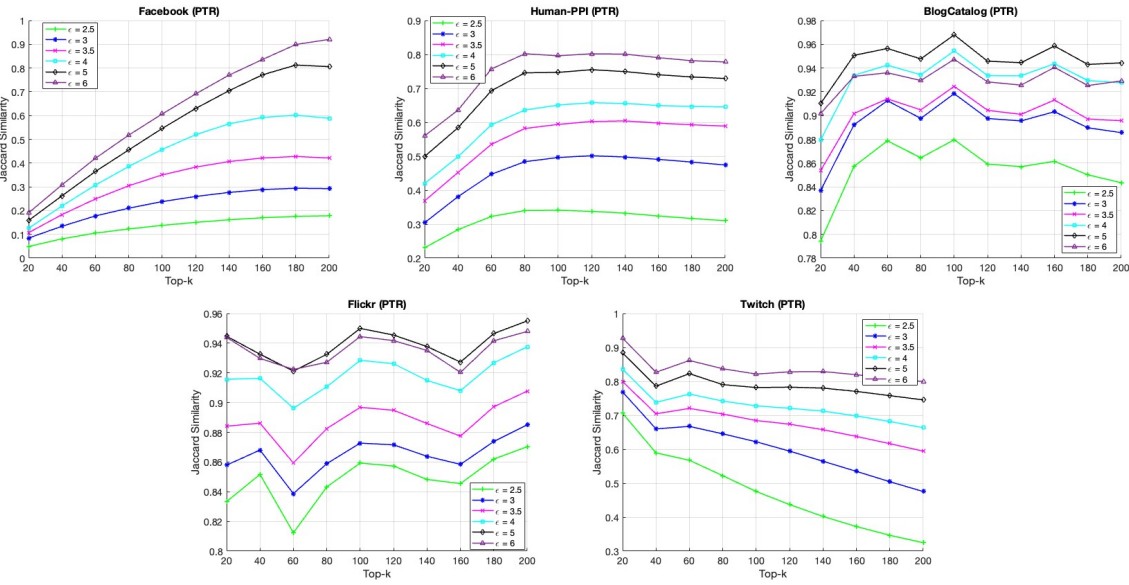

Figure 7: Jaccard similarity versus subset size ($k$) for Algorithm 1 under $(\epsilon_0, \delta_0) = (1, 7 \times 10^{-7})$ and varying $\epsilon_1 = \epsilon_2 = \epsilon$ values across real-world datasets. The privacy parameter is set to $\delta = \log(m)/m$, where $m$ denotes the number of edges in each network.

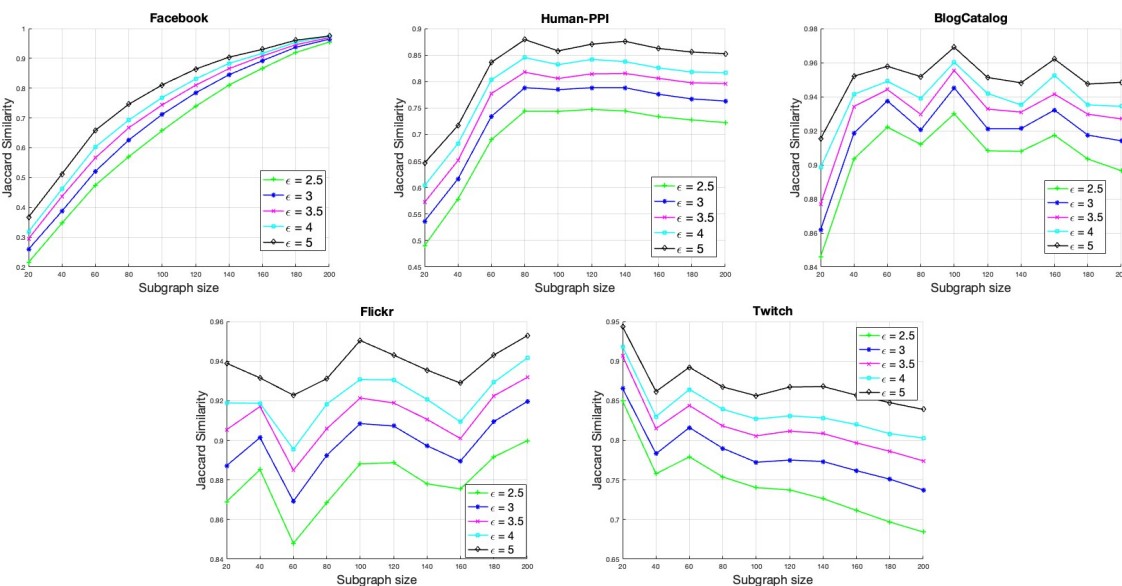

Figure 8: Jaccard similarity versus subset size ($k$) for Algorithm 2 under varying $\epsilon$ values with $\delta = \log(m)/m$ across real-world datasets.

