# OpenReview forum: "Differentially Private and Scalable Estimation of the Network Principal Component"
_TMLR — Accepted by TMLR_

### Review · Reviewer_xXLJ · 2025-12-11

**Summary Of Contributions:**

The paper studies edge-differentially private estimation of the principal eigenvector of a graph adjacency matrix and its use for two tasks: (A1) selecting top-k eigenvector-central nodes and (A2) approximating the densest-k-subgraph (DkS). The main technical contributions are:

A scalable Propose-Test-Release (PTR) variant for private PC: they construct a tractable sensitivity-1 surrogate ϕ(G), handle “large-gap” and “small-gap” regimes, and use the truncated biased Laplace mechanism to privately test whether the instance is well-behaved. The overall complexity is essentially that of computing a non-private PC.

An empirical comparison with the private power method (PPM) on several real graphs up to 3M nodes, showing that PTR achieves comparable utility to PPM, while being 2–3.5 orders of magnitude faster (Table 3, Figs. 2–3).

**Audience:**

Yes

**Audience Explanation:**

This paper sits at the intersection of several active areas:

Differential privacy for structured data: Edge-DP on graphs, especially at scale, is a central topic for privacy-preserving social network.

Spectral methods and graph mining: Eigenvector centrality and densest-subgraph-type problems are widely used in influence maximization, fraud detection, etc.

Readers interested in DP, graph algorithms will find the theoretical construction (PTR with a tractable surrogate) and the empirical evidence on million-node graphs relevant.

**Broader Impact Concerns:**

I do not see major negative broader-impact concerns.

**Claims And Evidence:**

Yes

**Claims Explanation:**

For the theoretical claims, the paper gives precise statements and sketches of the main arguments in the body, with detailed proofs deferred to the appendices.

While I did not check all proofs in full detail, the chain “local sensitivity → PTR surrogate → privacy guarantee → success probability” is logically consistent and uses standard DP tools, so the DP and sensitivity claims are reasonably convincing.

**Some concerns on empirical results.** The paper uses different $\delta$ for PTR and PPM, such as $\frac{1}{m}$ for PTR while $10^{-12}$ for PPM, this will make the utility favor PTR. Moreover, usually it requires $\delta$ at least $\frac{1}{m}\log m$ for $(\epsilon, \delta)$ DP.

**Requested Changes:**

Regarding the empirical results, please compare the algorithm under the same privacy budget, i.e., using the same and valid $\delta$s.

My main request is to explain the approach more accessibly and systematically in the main text, so that readers who are not already experts in PTR and sensitivity-based DP can follow the design choices.

High-level pipeline overview. Add a short “algorithm overview” subsection early. A small flow diagram or pseudo-code emphasizing data flow (what depends on the graph, where noise is added, what’s public) would help.

**Requested clarification on Eq. (7), Eq. (8), and Lemma 1**

One technical part that I found hard to follow is the derivation and role of Eq. (7). In the large-gap regime, you replace the constraint $LS_v(G') \ge \beta$ in (4) with $\theta(G') \ge \beta$ in (7). Some intuitive discussion is required on what $\theta(G')$ represents in both regimes (large-gap vs. small-gap) and how these choices of surrogate $\phi$ are used by PTR, which would make this part of the paper considerably easier to follow.

---

> ### Author Response · Authors · 2026-02-13
> **Response by authors**
>
> Thank you for your review. Please find our responses below.
>
> **(1) Empirical comparisons:** Following your suggestion, we re-performed all experiments with $\delta = \log(m)/m$ for both PPM and PTR, as requested. The qualitative trend of the plots remained unchanged, and we have updated the revised version of the paper with the new revised figures.
>
> **(2) Accessibility:** Thank you for the suggestion. We have added a flow-diagram of the proposed approach in Figure 2 and provided a high-level overview to make the presentation more accessible.
>
> **(3) Clarification of Eq. (7), Eq. (8), and Lemma~1:**
>
> *Large-gap regime:* ($\mathrm{GAP}(G) > t$): Please refer to our response to the third comment of reviewer GbSi. In this regime, Theorem (2) asserts that under assumptions (A1)-(A2), $\theta(G')$ serves as an upper bound on $LS_{v}(G')$.
>
> *Small-gap regime:* ($\mathrm{GAP}(G) \le t$).
> In this regime, assumption (A1) in Theorem 2 does not apply, and $\theta(G')$ is no longer guaranteed to be an upper bound on $LS_{v}(G')$, since bounding the local sensitivity is difficult.
> Hence, we want the PTR algorithm to stop w.h.p for such datasets. To this end, we deliberately choose $0$ to be the lower bound for the true distance $\gamma(G)$ in this regime. Using $0$ as the surrogate causes the test phase of PTR to stop w.p $1-\delta$, which is exactly the intended behavior for such ``unfavorable'' instances.
>
> Lemma 1 establishes that this value $0$ can be viewed as the solution of the optimization problem (8) where the r.h.s. of the constraint is $0$ instead of $\beta$. This form is useful because it allows a unified formulation (9) for the surrogate, which is applicable for both regimes.

---

### Review · Reviewer_GbSi · 2025-12-26

**Summary Of Contributions:**

The paper studies how to privately compute the top eigenvector (principal component) of a graph’s adjacency matrix under edge-DP and applied to tasks like eigenvector centrality and approximating densest-k-subgraph. The paper proposed an instance-specific Propose–Test–Release method that (i) privately tests whether the graph is “well-behaved” (using a truncated biased Laplace mechanism to avoid false positives), (ii) privately tests whether it’s safe to release, and then (iii) releases a one-shot noisy principal component with noise scaled to an upper bound on the local sensitivity. Experimental results, comparing to the iterative private power method baseline, shows that PTR can maintain comparable utility but achieve large speedups.

**Audience:**

Yes

**Audience Explanation:**

The paper leverages local sensitivity to reduce the noise scale and introduces a surrogate lower bound in the PTR test. This lower bound depends on the gap regime, and the required gap regime is also DP-guaranteed.

The method reduces PTR overhead to essentially computing a principal component plus a small number of private tests, enabling scalability to graphs with millions of nodes and large runtime improvements over iterative DP baselines.

It provides (to the best of my knowledge) the first edge-DP algorithm for DkS, which can potentially support a broader range of downstream tasks.

The resulting closed-form solution is elegant and easy to implement.

Overall, this is a solid paper with interesting results.

**Broader Impact Concerns:**

This is mainly a theory paper.  No obvious concerns on the ethical implications

**Claims And Evidence:**

Yes

**Claims Explanation:**

The claims are well supported by the theoretical analysis and the experiments show the improved performance over existing baselines. I went through the proofs briefly and did not find mistakes.

**Requested Changes:**

The empirical results appear not stable and general across datasets (e.g., Figure 3 for Facebook and Figure 2).

The writing could be more polished and consistent; for example, citation commands are sometimes used incorrectly (e.g., `\citet`, `\citealt`, `\cite`).

Please also check notation consistency, especially the use of \Phi(\cdot) and \Phi(\dot).

How does replacing the "true bad set" condition $L S_v\left(G^{\prime}\right) \geq \beta$ with the relaxed surrogate $\theta\left(G^{\prime}\right) \geq \beta$ guarantee that $\phi(G)$ is a valid lower bound on the true PTR distance $\gamma(G)$, and what intuition explains why this makes the PTR test conservative (i.e., less likely to mistakenly release)?

Why does using $\tilde{f}(G)=f(G)-\tilde{Z}$ with $\tilde{Z} \geq 0$ (TBL) eliminate false positives in the gap test compared to symmetric Laplace noise, and what is the tradeoff (in terms of false negatives / increased rejection probability) introduced by this one-sided, truncated noise?

---

> ### Author Response · Authors · 2026-02-13
> **Response by authors**
>
> Thank you for your review. Please see our responses below.
>
> **(1) Empirical results:** Both Figures 2–3 already average across 200 Monte-Carlo trials, and exhibit small variance around the average value, indicating both consistency and reliability. Moreover, the qualitative behavior is consistent across datasets - PTR and PPM closely track the non-private baseline in Figure 3 and exhibit similar solution quality for top-$k$ subset estimation. This pattern holds across all six graphs, despite large differences in size and structure. For additional experiments, please refer to Figures 5,6,7,8 in the Appendix.
>
> **(2) Writing and notation consistency:** Thank you for your careful reading. We have made these changes.
>
> **(3) On replacing true condition with relaxed surrogate:** Theorem $2$ asserts that under assumptions (A1)-(A2), it holds that $\theta(G') \geq LS_{v}(G')$. We exploit his fact by replacing the constraint
> $LS_{v}(G') \ge \beta$
> in problem (4) with
> $\theta(G') \ge \beta,$
> which results in a larger feasible set. Minimizing the distance $d(G,G')$ over a larger feasible set can only decrease the optimal value $\phi(G)$. This yields $\phi(G) \le \gamma(G)$, which is precisely the requirement
> $
> \gamma(G) \ge \phi(G) \ge 0
> $
> used by the modified PTR framework of Li et al. (2024).
>
> **(4) On conservativeness:** The test phase of the standard PTR algorithm adds noise to the true distance $\gamma(G)$ and then checks if it exceeds a noisy threshold before releasing an output. In our case, we add noise to the surrogate $\phi(G)$ instead, which is a lower bound on $\gamma(G)$. Hence, the noisy lower bound $\hat{\phi}(G)$ can be smaller compared
> to its counterpart computed based on $\gamma(G)$. This makes it harder for $\hat{\phi}(G)$ to exceed the release threshold, leading to more rejections rather than unsafe releases. In other words, the relaxation can introduce false negatives (rejecting even when the true $\gamma(G)$ is large), but it does not create false ``safe'' certifications of distance.
>
> **(5) On the Truncated Biased Laplace Mechanism (TBLM):** The TBLM is used to privatize the function
> $f(G) = \mathrm{GAP}(G) - t$, where
> $t = 2(\sqrt{2}+1).$
> The aim is to test (in a differentially private manner) whether the graph $G$ belongs to the small or large gap-regime.
> If $f(G) \le 0$, we want the algorithm to stop w.h.p. and to avoid incorrectly treating the instance as ``large-gap'' (a false positive), because that routes the algorithm to the large-gap surrogate $\phi(G)$ and downstream release logic.
>
> Adding two-sided Laplace noise $Z$, we obtain
> $
> \tilde{f}(G) = f(G) + Z.
> $
> However, note when $f(G)$ is negative, $\tilde{f}(G)$ can be positive with nontrivial probability, leading to false positives.
>
> We instead sample $\tilde{Z} \ge 0$ from the TBL distribution and set
> $
> \tilde{f}(G) \;=\; f(G) - \tilde{Z}.
> $
> Since we only subtract nonnegative noise, we have $\tilde{f}(G) \le f(G)$. Therefore, if $f(G) \le 0$, then $\tilde{f}(G) \le 0$ always. False positives are eliminated by construction.
>
> The fact that we are using one-sided noise makes the test more conservative. While false positives cannot occur,
> it can introduce additional false negatives - consider a graph for which $f(G) > 0$, but is close to $0$. Then, subtracting $\tilde{Z}$ may yield $\tilde{f}(G) < 0$, incorrectly classifying a large-gap instance as small-gap. This then sets
> $
> \phi(G) = 0
> $
> and causes the algorithm to stop with high probability. Overall, this increases rejection probability for graphs whose gap is above, but near the threshold $t$, but protects against unsafe regime mis-classification.

---

### Review · Reviewer_zSQ1 · 2026-02-01

**Summary Of Contributions:**

This paper studies the problem of DP computation of the principal eigenvector of a graph adjacency matrix under edge privacy. It first analyzes why existing instance-specific mechanisms based on smooth sensitivity fail to significantly improve utility. It shows in this setting, smooth sensitivity bounds remain close to the global sensitivity despite large spectral gaps in graphs. Motivated by this limitation, the paper develops a practical and scalable implementation of the Propose-Test-Release framework tailored to principal component estimation. By deriving tractable sensitivity surrogates and closed-form parameter selection rules, the proposed method achieves one-shot noise addition with computational complexity comparable to non-private eigenvector computation. The resulting algorithm is applied to private eigenvector centrality extraction and private approximation of the densest k-subgraph problem.

**Additional Comments:**

Aside from the weaknesses stated in the Requested Changes section, this paper has several strengths. The paper is well written and easy to follow, with a clear logical structure. Although the contributions are incremental and confined to a specific setting, the connections to and motivations from existing work are well constructed and clearly articulated. In addition, the theoretical quantifications provided throughout the paper are helpful for understanding the paper. In particular, the paper offers a convincing analysis explaining why global sensitivity based output perturbation and smooth sensitivity fail to provide meaningful utility improvements for graph principal component estimation.

**Audience:**

Yes

**Audience Explanation:**

Privacy and graph related topics are popular among machine learning and signal processing communities.

**Claims And Evidence:**

Yes

**Claims Explanation:**

The paper clearly states their contributions and relevance to the exitisng works.

**Requested Changes:**

1. My main technical concern is the reliance on the spectral gap assumption. The proposed method depends critically on the presence of a sufficiently large eigen-gap, and its behavior outside this regime is not well characterized. The paper should more clearly identify classes of graphs for which this assumption is likely to fail and discuss the expected performance in such settings. If a quantitative characterization is not feasible, a qualitative discussion would still be valuable.

2. Additionally, I am also concerned about the frequency with which the PTR mechanism returns "no response" across different datasets and privacy budgets, and how such outcomes impact downstream applications. Reporting empirical rejection rates or providing guidance on handling these cases would strengthen the practical relevance of the method.

---

> ### Author Response · Authors · 2026-02-13
> **Response by authors**
>
> Thank you for your review. Please find our responses below.
>
> **(1) On the gap assumption:** We have provided a minimum threshold on the spectral gap equal to $2(\sqrt{2+1})$ (which is a universal constant) in [Theorem 2, (A1)]. As explained in the proof of Theorem 3, Appendix H, for all datasets with gap below this threshold, the algorithm refuses to release an output for such datasets w.p. at least $1-\delta$. We have emphasised this point in the main body of the paper.
>
> For graphs whose gap exceed this threshold, characterizing the performance is more challenging. This is because the added noise is dependent not only on the gap, but also on the spread of the energy of the entries in the leading eigen-vector. From equation (19), it can be seen that for a fixed privacy budget and other algorithm parameters, the proposed bound $\beta$ behaves like
> $$\beta \approx O\biggl(\frac{\sqrt{v_{\pi(1)}^2 + v_{\pi(2)}^2}}{GAP(G)}\biggr),$$
> where $v_{\pi(1)},v_{\pi(2)}$ are the two largest entries of the eigen-vector $\mathbf{v}$. This is in line with the local sensitivity bound derived in Theorem 1. Hence, $\beta$ being small depends on the gap being large, *and* the ``energy spread'' in the entries of $\mathbf{v}$ being small. We have made this point more clear in the revised version.
>
> While its difficult to answer for which classes of graphs these conditions are not jointly fulfilled, it turns out that there is a precise answer to the complementary question -  specifically, the class of expander graphs. Following the terminology of [R1], we designate a graph $G$ as being an $(n,d,\alpha)$-expander if it fulfills the following conditions.
>
> (1) $G$ has $n$ vertices and each vertex has degree $d$, i.e., $G$ is $d$-regular.
>
> (2) The second largest eigen-value of the adjacency matrix (in magnitude) is $|\lambda_2| \leq \alpha d$, where $\alpha \in (0,1)$, and $1-\alpha$ represents the expansion coefficient. Smaller values of $\alpha$ correspond to higher expansion.
>
> For such a class of graphs, the following facts are known.
>
> [F1] The principal eigen-vector $\mathbf{v} = \frac{1}{\sqrt{n}}\mathbf{1}_n$.
>
> [F2] The largest eigen-value of the adjacency matrix of $G$ is $\lambda_1 = d$.
>
> [F3] For fixed $d$, as $n \rightarrow \infty$ it holds that $|\lambda_2| \geq 2\sqrt{d-1} - o_n(1)$
>
> From these facts, we obtain that (a): $v_{\pi(1)}^{2} = v_{\pi(2)}^{2} = 1/n$, i.e., the entries of $\mathbf{v}$ are uniformly spread out in terms of energy. In addition,
> (b): the spectral gap
> $GAP(G) = \lambda_1 - |\lambda_2|$ satisfies
> $$d(1-\alpha) \leq GAP(G) \leq d - 2\sqrt{d-1}+ o_n(1)$$
> Hence, the main figure of merit $\frac{\sqrt{v_{\pi(1)}^2 + v_{\pi(2)}^2}}{GAP(G)}$ can be sandwiched as
> $$\frac{\sqrt{2}}{\sqrt{n}[2\sqrt{d-1}+ o_n(1)]} \leq
>     \frac{\sqrt{v_{\pi(1)}^2 + v_{\pi(2)}^2}}{GAP(G)} \leq \frac{\sqrt{2}}{\sqrt{n}d(1-\alpha)}
> $$
> Suppose that $d(1-\alpha)$ exceeds the threshold $t$ - a calculation reveals that $d \geq 12$ is sufficient for every admissible $\alpha \in (0,1)$. As a consequence, $GAP(G) \geq t$.
> Then, for a large $n$ and a fixed $d$, $\beta = \Theta(1/\sqrt{n})$ and the algorithm will release outputs with a small level of noise. Although real-world graphs do not conform precisely to such a model, empirical studies reveal that they can possess good expansion properties (e.g., [R2]), which makes them a good candidate for our algorithm.
>
> [R1]: S. Hoory, N. Linial, and A. Wigderson. ``Expander graphs and their applications.'' Bulletin of the American Mathematical Society 43, no. 4 (2006): 439-561.
>
> [R2]: F. D. Malliaros and V. Megalooikonomou. ``Expansion Properties of Large Social Graphs''.
> International Conference on Database Systems for Advanced Applications. Berlin, Heidelberg: Springer Berlin Heidelberg, 2011.
>
> **(2) Handling No response:** We thank the reviewer for raising this point. A "no response" is due to (a) the dataset not clearing the private gap test or (b) the dataset clearing the gap test but failing the final release step. For the latter case, Theorem 5 provides an explicit lower bound on the success probability, which allows end-users to directly control the probability of successful private release. In all experiments, the datasets we used successfully cleared the gap test, and we set the probability of release to be at least 0.95. We empirically observed that PTR returns a valid output in the vast majority of trial runs across all datasets (see added Table 4 in Section 8 in the revised version.)
>
> For other datasets in the wild, if a "no response" is obtained, one could repeat the PTR mechanism (since it is very fast) or increase the lower bound on the success probability to improve the odds of a response. However, this also increases $\beta$ and results in addition of greater levels of noise to the principal component. If the mechanism repeatedly returns ``no response'', then one could fall back to using PPM. We have added these discussions in Section 5 (selection of $\beta$) and 8.1.

---

### Author Response · Authors · 2026-02-13
**Response to Editor and Reviewers**

We sincerely thank you for your valuable comments and suggestions on our manuscript. We have carefully addressed all the points raised and have uploaded the revised version of the manuscript for your consideration (all changes are marked in blue). We greatly appreciate your time and effort in reviewing our work.

---

### Decision · Action_Editor_1CB1 · 2026-03-27

**Recommendation:** Accept as is

**Audience:**

Yes

**Audience Explanation:**

Differentially private estimation of graph statistics is of interest to several subcommunities within the broader TMLR audience.

**Claims And Evidence:**

Yes

**Claims Explanation:**

The main theoretical claims are that the proposed method satisfies differential privacy and achieves a specified utility guarantee.  No reviewers flagged any potential issues with the proofs of these claims.  The paper also empirically evaluates the method relative to a specific baseline implementaiton of the private power method and the claims made about that evaluation are supported in the paper.